EMBO
Molecular Medicine

# CDK4 phosphorylation status and a linked gene expression profile predict sensitivity to palbociclib

Eric Raspé[1,2,*] (ID), Katia Coulonval[1,2,†], Jaime M Pita[1,2,†], Sabine Paternot[1,2,†], Françoise Rothé[2,3,†], Laure Twyffels[4,5], Sylvain Brohée[2,3], Ligia Craciun[6], Denis Larsimont[7], Véronique Kruys[4,5], Flavienne Sandras[8,9], Isabelle Salmon[2,8,9], Steven Van Laere[10], Martine Piccart[2,11], Michail Ignatiadis[2,3], Christos Sotiriou[2,3,**] (ID) & Pierre P Roger[1,2,***] (ID)

## Abstract

Cyclin D-CDK4/6 are the first CDK complexes to be activated in the G1 phase in response to oncogenic pathways. The specific CDK4/6 inhibitor PD0332991 (palbociclib) was recently approved by the FDA and EMA for treatment of advanced ER-positive breast tumors. Unfortunately, no reliable predictive tools are available for identifying potentially responsive or insensitive tumors. We had shown that the activating T172 phosphorylation of CDK4 is the central rate-limiting event that initiates the cell cycle decision and signals the presence of active CDK4. Here, we report that the profile of post-translational modification including T172 phosphorylation of CDK4 differs among breast tumors and associates with their subtypes and risk. A gene expression signature faithfully predicted CDK4 modification profiles in tumors and cell lines. Moreover, in breast cancer cell lines, the CDK4 T172 phosphorylation best correlated with sensitivity to PD0332991. This gene expression signature identifies tumors that are unlikely to respond to CDK4/6 inhibitors and could help to select a subset of patients with HER2-positive and basal-like tumors for clinical studies on this class of drugs.

**Keywords** CDK4/6 inhibitors; breast cancer; cyclin-dependent kinase 4; PD0332991; sensitivity biomarker

**Subject Categories** Cancer; Chromatin, Epigenetics, Genomics & Functional Genomics; Pharmacology & Drug Discovery

## Introduction

Deregulation of the cell division cycle necessarily affects all cancers (Sherr & McCormick, 2002; Malumbres & Barbacid, 2009; Hanahan & Weinberg, 2011). Cell cycle commitment at the G1 phase restriction (R) point is initiated by inactivating phosphorylations of the central cell cycle/tumor suppressor pRb by the cyclin-dependent kinases CDK4 and CDK6. These kinases are activated by D-type cyclins upon mitogenic/oncogenic signaling (Sherr, 1995; Bartek et al, 1996; Bockstaele et al, 2006a; Asghar et al, 2015; Sherr et al, 2016). pRb phosphorylation in committed cells is maintained by a positive feedback loop linking pRb to E2F-dependent transcription of cyclin E, which activates CDK2 and leads to further phosphorylation of pRb (Lundberg & Weinberg, 1998). This feedback is completed by the self-induction of E2F1 and by the mutual inhibition between cyclin E-CDK2 and p27[Kip1]. Together, these events convert graded mitogen inputs into all-or-none E2F responses and cell cycle commitment (Yao et al, 2008). CDK4 activity requires its binding to cyclins D (CCND1-3 genes). INK4 CDK4 inhibitors such as p16 (CDKN2A-D genes) compete for this binding (Sherr, 1996; Asghar et al, 2015). Importantly, CDK4 activation also requires its phosphorylation at T172 (Kato et al, 1994a,b). This activation step has been much less studied due to lack of easy detection tools (Bockstaele et al, 2006a). By separating the modified forms of CDK4 by two-dimensional (2D) gel electrophoresis, we have shown that the activating T172-phosphorylation of CDK4 bound to cyclin D is the central rate-limiting event in CDK4 activation. This event determines pRb phosphorylation and cell cycle commitment in

1 WELBIO and Institute of Interdisciplinary Research (IRIBHM), Campus Erasme, Université Libre de Bruxelles (ULB), Brussels, Belgium
2 ULB-Cancer Research Center (U-CRC), Université Libre de Bruxelles, Brussels, Belgium
3 Breast Cancer Translational Research Laboratory, Institut Jules Bordet, Université Libre de Bruxelles (ULB), Brussels, Belgium
4 Laboratoire de Biologie Moléculaire du Gène, Faculté des Sciences, Université libre de Bruxelles (ULB), Brussels, Belgium
5 Center for Microscopy and Molecular Imaging, Université Libre de Bruxelles (ULB), Brussels, Belgium
6 Tumor Bank of the Jules Bordet Institute, Université Libre de Bruxelles (ULB), Brussels, Belgium
7 Department of Pathology, Institut Jules Bordet, Université Libre de Bruxelles (ULB), Brussels, Belgium
8 Department of Pathology, Erasme Hospital, Université Libre de Bruxelles (ULB), Brussels, Belgium
9 Biobank of the Erasme Hospital, Université Libre de Bruxelles (ULB), Brussels, Belgium
10 Center for Oncological Research (CORE), University of Antwerp, Antwerp, Belgium
11 Medical Oncology Clinic, Department of Medicine, Institut Jules Bordet, Université Libre de Bruxelles (ULB), Brussels, Belgium
*Corresponding author. Tel: +32 2 555 41 53; E-mail: eraspe@ulb.ac.be
**Corresponding author. Tel: +32 2 541 34 28; E-mail: christos.sotiriou@bordet.be
***Corresponding author. Tel: +32 2 555 41 53; Fax: +32 2 555 46 55; E-mail: proger@ulb.ac.be
†K. Coulonval, J.M. Pita, S. Paternot and F. Rothé had main contributions by acquiring essential sets of experimental data

 

pRb-proficient cells (Bockstaele *et al*, 2006a; Paternot *et al*, 2010; Bisteau *et al*, 2013). T172 phosphorylation of CDK4 is exquisitely regulated in various cell models and mitogenic regulations (Kato *et al*, 1994a; Paternot *et al*, 2003, 2010; Bockstaele *et al*, 2006b, 2009; Rocha *et al*, 2008; Paternot & Roger, 2009; Blancquaert *et al*, 2010; Bisteau *et al*, 2013; Merzel Schachter *et al*, 2013), whereas T177 phosphorylation of CDK6 was found to be weak and unregulated or absent (Bockstaele *et al*, 2006b, 2009). Moreover, in contrast to CDK2 and CDK1, CDK4 activation is not restricted by stoichiometric inhibitory phosphorylations (Kato *et al*, 1994b; Bockstaele *et al*, 2006a,b).

One in eight women is diagnosed with breast cancer, a heterogeneous disease with variable histology, clinical presentation, and response to therapy. Prognosis and treatment of breast cancer are significantly informed by biomarkers. For example, estrogen receptor alpha (ER)-positive status, which is detected in 70% of breast cancers, predicts a more favorable outcome and indicates treatment with endocrine therapy. ER-negative/HER2-positive status is associated with a less favorable outcome, though it has been considerably improved by targeted therapies such as anti-HER2 antibodies. In contrast, tumors lacking estrogen, progesterone, and HER2 receptors (triple-negative tumors) have the worst outcome and are treated mainly by genotoxic chemotherapy.

The use of gene expression profiles has further classified breast tumors into at least five biologically and clinically relevant molecular subtypes: the low proliferative ER-positive/HER2-negative luminal A, the high proliferative ER-positive/HER2-negative luminal B, the basal-like (mainly triple-negative), the HER2-enriched, and the normal-like subtypes (Sorlie *et al*, 2001; Sotiriou *et al*, 2006; Cleator *et al*, 2007; Prat & Perou, 2011; Prat *et al*, 2012a). In breast tumors, the alterations of the CDK4/pRb axis include amplification of *CCND1* or *CDK4*, loss of *CDKN2A/B* or, less often, loss or mutation of pRb (Ertel *et al*, 2010; Curtis *et al*, 2012; The Cancer Genome Atlas Network, 2012). Preclinical studies using breast tumor cell lines (Finn *et al*, 2009; Dean *et al*, 2010; Miller *et al*, 2011), mouse models (Choi *et al*, 2012), and *ex vivo* patient tumor cells (Dean *et al*, 2012) have demonstrated the efficacy of inhibiting CDK4/6 by PD0332991 (palbociclib) to arrest proliferation. Resistance to this drug is mainly ascribed to pRb loss and *CCNE1* amplification (Wang *et al*, 2007; Dean *et al*, 2012; Herrera-Abreu *et al*, 2016). PD0332991 and other CDK4/6 inhibitory drugs (LEE-011, Novartis; LY2835219, Eli Lilly) are being tested in a growing number of phase II/III clinical trials against most cancers (91 studies totaling 15,500 patients recorded in *ClinicalTrials.gov*; Asghar *et al*, 2015; Hamilton & Infante, 2016; Patnaik *et al*, 2016; Sherr *et al*, 2016). In multiple breast cancer clinical trials (Asghar *et al*, 2015; DeMichele *et al*, 2015; Finn *et al*, 2015, 2016; Cristofanilli *et al*, 2016; Hamilton & Infante, 2016; Sherr *et al*, 2016), PD0332991 combined with endocrine treatment substantially improved progression-free survival compared to endocrine treatment alone in women with ER-positive/HER2-negative advanced breast cancer. Therefore, PD0332991 in combination with endocrine therapy was approved by the FDA (February 2015) and EMA (November 2016) as a first-line treatment for advanced ER-positive breast cancers.

Except the currently used estrogen receptor status of breast cancers, no reliable biomarkers could be defined to diagnose tumors that depend on CDK4 activity and hence would respond to CDK4/6 inhibitors (Dickson, 2014; Asghar *et al*, 2015; DeMichele *et al*, 2015; Sherr *et al*, 2016). Here, we show that the presence or absence of the T172-phosphorylated CDK4 form and its relative abundance varies among breast tumors according to their molecular subtypes and risk and that it predicts the response of breast cancer cell lines to PD0332991. To overcome the difficulty of using proteomic analysis in the clinic, we developed a surrogate CDK4 modification signature based on the expression of 11 genes. This signature correctly predicts the CDK4 modification profile of tumors and breast cancer cell lines, and the sensitivity of the latter to PD0332991. This signature identifies tumors that are likely to be insensitive to CDK4 inhibitors. Once adapted into a clinically validated assay, it may optimize the use of the drug and extend its indication to most HER2-positive and some basal-like tumors.

## Results

### CDK4 modification profile varies in breast tumors and is associated with specific molecular subtypes

The presence of phosphorylated active CDK4 has never been assessed in tumors. Detection of CDK4 phosphorylation is complicated by the lack of adequate antibodies, the absence of a SDS–PAGE migration shift associated with phosphorylation, and the very low expression level of CDK4. Therefore, we modified our 2D-gel electrophoresis immunodetection assay to analyze the modified and native CDK4 forms in minimal amounts of frozen breast tumor samples extracted in urea buffer. As previously characterized (Bockstaele *et al*, 2006b), CDK4 was resolved by its charge into three main forms in breast cancer cell lines such as MCF7 cells and in breast tumor samples (Fig 1A–D). The most basic form (form 1) was the native CDK4. The most acidic form (form 3), which increased in response to proliferation stimulation of MCF7 cells (Fig 1B), had been identified as the highly regulated T172-phosphorylated CDK4 form using several approaches (Bockstaele *et al*, 2006b) including a T172-phosphospecific antibody (Fig 1A and C). Another yet unidentified modified CDK4 form (form 2), which is variably observed in tissue samples and in most cell lines, does not incorporate [$^{32}$P] phosphate (Bockstaele *et al*, 2006b).

The relative abundance of these three CDK4 forms was compared in normal breast tissue obtained from reduction surgery and in an exploratory set of 19 breast tumors for which clinical records and gene expression microarray data are available (accession GSE20713; Dedeurwaerder *et al*, 2011; Fig 1D). In breast tumor samples, the relative abundance of the T172-phosphorylated form 3 of CDK4 was highly variable. We decided to class the profiles of CDK4 modifications in three categories. In a subset of nine tumors, both the T172-phosphorylated form 3 and the intermediate CDK4 form 2 were detected. In this profile (profile H for *high*), the abundance of the phosphorylated form was greater than or almost equal to the abundance of the intermediate form 2. In a second group of six tumors, the phosphorylated form was also detected, but its abundance was lower than that of the intermediate form 2 (profile L for *low*). Interestingly, these profile L tumors had lower Ki-67 labeling (Fig 1E). Finally, the T172-phosphorylated CDK4 was undetectable in four tumors (profile A for *absent*). These tumors paradoxically presented the highest Ki-67 labeling rates (Fig 1E). Hence, this

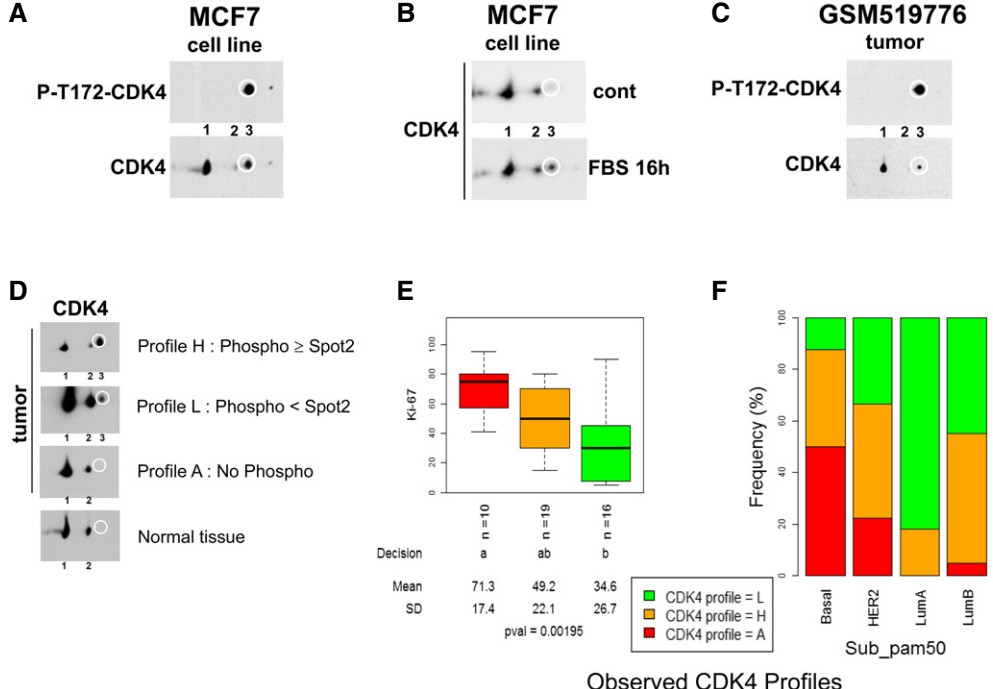

**Figure 1. CDK4 modification profiles in breast cell lines and tumors and their relations to the tumor Ki-67 labeling index and molecular subtypes.**

A–C   Identification of the CDK4 T172-phosphorylated form after separation by 2D-gel electrophoresis of protein extracts and immunodetection with a CDK4 T172-phosphospecific antibody (P-T172-CDK4) or the CDK4 H22 antibody. The main T172-phosphorylated form (spot 3) is circled. Extracts were from (A) asynchronous MCF7 cell line, (B) MCF7 cells rendered quiescent by serum withdrawal and addition of fulvestrant (cont) or re-stimulated for 16 h with FBS, or (C) frozen sections of a breast tumor sample.

D   Proteins were extracted from frozen sections of breast tumor samples or normal tissue, resolved by 2D-gel electrophoresis and detected with the CDK4 H22 antibody. Three representative modification profiles of CDK4 are shown. The position of the T172-phosphorylated CDK4 form (spot 3) is circled. Profile H indicates strong relative presence of the T172-phosphorylated CDK4 form 3 vs. modified (non-phosphorylated) form 2. Profile L indicates weak presence of the phosphorylated CDK4 form 3 vs. form 2. Profile A indicates absence of phosphorylated CDK4.

E   The Ki-67 labeling index was determined in parallel FFPE sections of the tumors stained with the DAKO Ki-67 antibody. Data (box and whiskers) represent median, quartiles and the largest and smallest values. Pairwise comparisons were performed with the Kruskal–Wallis test (level of confidence set at 0.95). The first line below the plot indicates the number of observations for each CDK4 modification profile (Ki-67 index was available for 45 of the 56 tumors in this study). The second line reports whether the true effect of observed CDK4 modification profile was considered significant (levels with the same letter are not significantly different at alpha set to 0.05). The third and fourth lines report the respective mean and SD. The last line provides the P-value of rejection of the null hypothesis that all means are equal.

F   The relative proportion of the three CDK4 modification profiles in the different intrinsic molecular subtypes is shown for the 56 breast tumor samples analyzed in this study. The molecular subtypes were defined with the genefu package based on expression values of the PAM50 selected genes in each tumor.

absence of CDK4 phosphorylation could not have been due to low proliferation rate. In normal breast (mostly quiescent tissue), only the native unmodified form 1 and the intermediate CDK4 form 2 were detected (Fig 1D, lowest profile).

The same profiles were observed in validation cohorts with quantifiable CDK4 profiles and at least 30% cellularity. These cohorts included 31 new untreated breast tumors from Jules Bordet Institute (Brussels) with newly acquired gene expression profiles (accession GSE87007) and six non-inflammatory breast tumors analyzed at the University of Antwerp (E-MTAB-1006) (Van Laere et al, 2007). Datasets EV1 and EV2 describe the demographic distributions of the clinical parameters in all the tumors analyzed in this study. The relative levels of the CDK4 forms 3 and 2 (Spot3/Spot2 ratios) were quantified using ImageJ. Two empirical thresholds of the Spot3/Spot2 ratio were used to define the three CDK4 modification profiles: profile A, Spot3/Spot2 ratio below 0.1; profile H, ratio ≥ 0.9; profile L, ratio 0.1–0.9. In the merged breast tumor cohorts, the relative proportions of tumors with profiles A, H and L

(Fig 1F) differed significantly (P-value = 0.0014 by chi-square test) between basal-like, HER2-positive, luminal A, and luminal B molecular subtypes as defined by the PAM50 gene expression classification (Parker et al, 2009; Prat et al, 2012b). Profile A tumors were enriched in basal-like tumors (8 of 16). Profile H tumors were more frequent among luminal B and HER2-positive tumors. Profile L tumors were most frequent among luminal A tumors. Remarkably, almost half of basal-like tumors displayed profile H (7 of 16). The relative proportions of these three CDK4 modification profiles among breast tumors were also significantly associated with their grade, estrogen receptor, progesterone receptor, and HER2 and triple-negative statuses, as well as with their genomic grade index (GGI) (Sotiriou et al, 2006) and Oncotype DX risk score (Paik et al, 2004) (upper panels in Fig EV1A and B, Dataset EV3). Importantly, most grade 1 tumors and tumors with low GGI or Oncotype DX risks displayed profile L. Profiles H and A tumors were enriched in grade 3 and high GGI tumors. Profile A tumors were also enriched in triple-negative tumors.

## Development of a surrogate marker of tumor sensitivity to CDK4 inhibitors based on correlation with CDK4 modification profiles

As tumors lacking active phosphorylated CDK4, the main target of CDK4/6 inhibitors, will likely be insensitive to these drugs, analysis of the CDK4 modification state may be clinically useful. Unfortunately, preservation and detection of the phosphorylation of a low abundance protein in formalin-fixed (FFPE) material is technically challenging. We therefore explored whether a gene expression profile could serve as a surrogate assay that faithfully predicts tumor CDK4 modification profiles and hence responsiveness to CDK4 inhibitors. The CDK4 modification profiles were used as categorical variables to compare the variations of the expression of specific genes or reported gene expression signatures among tumors. The GGI index, the Rb LOH score defined in Perou's laboratory (Herschkowitz *et al*, 2008), and the Rb loss index introduced by Knudsen's laboratory (Ertel *et al*, 2010) were significantly lower only in tumors with profile L (Fig 2A–C, Dataset EV4). This indicates that these scores cannot be used to predict

the absence of active phosphorylated CDK4. Moreover, the CDK4 modification profile could not be predicted by the expression of any single-cell cycle marker (Fig EV2, Dataset EV5). In most profile A tumors compared to profile H and L tumors, the expression levels of *CDKN2A* and *CCNE1* were specifically elevated and the expression levels of *RB1* and *CCND1* were lower. However, these values were not strictly related to each other or to the CDK4 modification profile (Fig EV3). For instance, two of the eleven tumors lacking CDK4 phosphorylation (profile A; one HER2-positive and one basal-like) displayed a high expression level of *CCNE1* but not of *CDKN2A* (Fig EV3).

Therefore, we investigated whether the combined expression of selected genes could be used to build a surrogate marker to predict the CDK4 modification profile of a tumor. Our supervised strategy was inspired by the one adopted in Perou's laboratory for defining the molecular subtypes of breast tumors (Sorlie *et al*, 2001). As detailed in the Appendix Supplementary Text, we used significance analysis of microarrays (SAM) to identify genes that were either differentially expressed among the three CDK4 modification profiles or

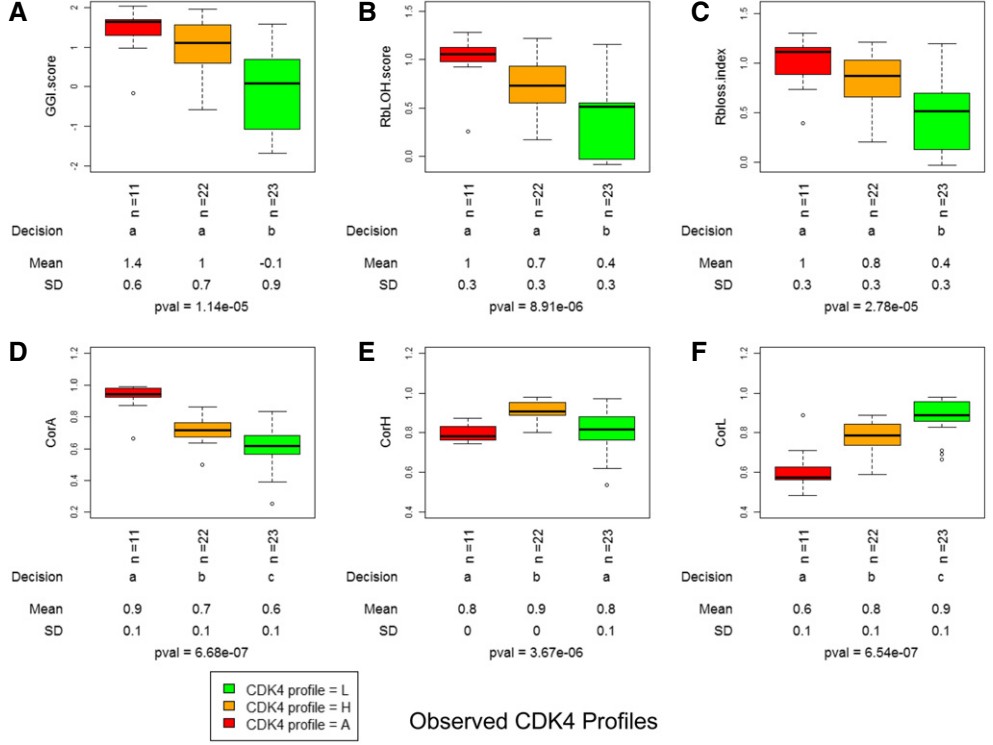

**Figure 2. Association of the three observed CDK4 modification profiles with the predictive scores of risk and Rb loss (A–C) or with the Spearman coefficient of correlation between the expression profiles of 11 genes and the corresponding references representative of the three CDK4 modification profiles (D–F).**

Data (box and whiskers) represent median, quartiles and the largest and smallest values with outliers excepted. The first line below the plots indicates the number of observations. The second lines reports whether the true effect of CDK4 modification profile is significant (levels with the same letter are not significantly different at alpha set to 0.05). The third and fourth lines report the respective means and SD. The last line provides the *P*-value of rejection of the null hypothesis that all means are equal. Pairwise comparisons were performed with the Kruskal–Wallis test (level of confidence set at 0.95).

A–C    GGI risk score (Sotiriou *et al*, 2006) (A), Rb LOH score (Herschkowitz *et al*, 2008) (B), and Rb loss index (Ertel *et al*, 2010) (C) were computed from the expression levels measured by the probe sets corresponding to the genes selected in the corresponding publications.

D–F    The gene expression levels of the 11 selected genes were measured in each tumor with the Affymetrix HG-U133 plus2 platform using the probe sets mentioned in Dataset EV7. These values were compared by Spearman correlation to three reference centroids representative of each of the three tumor CDK4 modification profiles. CorA (D), CorH (E), and CorL (F) indicate the coefficients of correlation to the reference centroids corresponding to the CDK4 modification profiles A, H, and L, respectively.

were associated with them. Probe selection was repeated for 50 randomly selected subsets of the tumors from the Bordet Institute as described in details in the Appendix Supplementary Text. Next, for each list of selected genes, we built three reference centroids by computing the mean expression value of each selected gene in tumors representative of each of the three CDK4 modification profiles. Centroids were defined within the same subset of tumors used to select probes. Finally, the gene expression profile of the selected genes in a particular tumor was compared to these three references by Spearman correlation. The predicted profile was the one corresponding to the centroid with the highest correlation coefficient. The performance of the classification was evaluated by comparing the proportions of matching observed/predicted profiles in the complementary subset of patients. The selected gene lists were optimized by stepwise removal of the probe contributing the least to the classification performance (see Appendix Supplementary Text for details). At the end of this procedure, a gene expression signature consisting of 11 genes (Fig 3, Dataset EV6) was selected as the shortest list providing the best agreement between observed and predicted CDK4 modification profiles in the three merged cohorts of breast tumors (84% agreement in the 56 tumors; all mismatches except one were between profiles H and L). Taken individually, with the exception of *CCNE1*, none of the 11 the genes clearly distinguished the three profiles (Appendix Fig S1). By contrast, the

correlation coefficients to the three profile references varied differently between tumors with profiles A, H, or L. Correlation coefficients to profile A reference decreased from profile A tumors to profile L ones (Fig 2D). The opposite was observed with correlation to profile L (Fig 2F). Correlation coefficients to profile H were only higher in profile H tumors (Fig 2E). The concordance rate between observed and predicted CDK4 modification profiles was reproduced neither after prediction based on 1,000 random lists of 11 genes nor after 1,000 random permutations of the patient labels (Appendix Fig S2). Hierarchical clustering of the expression levels of the 11 probes (Fig 3) correctly segregated profile A tumors from those with profile H or L, whereas Rb loss index and molecular subtype did not (Fig 3). A complementary relationship between the tumor Ki-67 labeling index and the expression of the 11 genes was noted (Appendix Fig S3). Expression of *CCDC99*, *NUP155*, *TAGLN2*, and to some extent *TIMM17A* and *CCNE1* was positively correlated to the Ki-67 labeling index. Expression of *FBXL5*, *TP53TG1*, *PPP1R3C*, and to a lesser extent *RAB31* and *GSN* was negatively correlated to the proliferation index. Similar correlations of the expression levels of these 11 genes to the gene expression of proliferation markers were observed (Dataset EV7). The coefficients of correlation to the references of CDK4 profiles A and H were positively correlated to GGI, Oncotype Dx, and Rb LOH scores as well as to the Ki-67 labeling index (Appendix Fig S4). In contrast, the

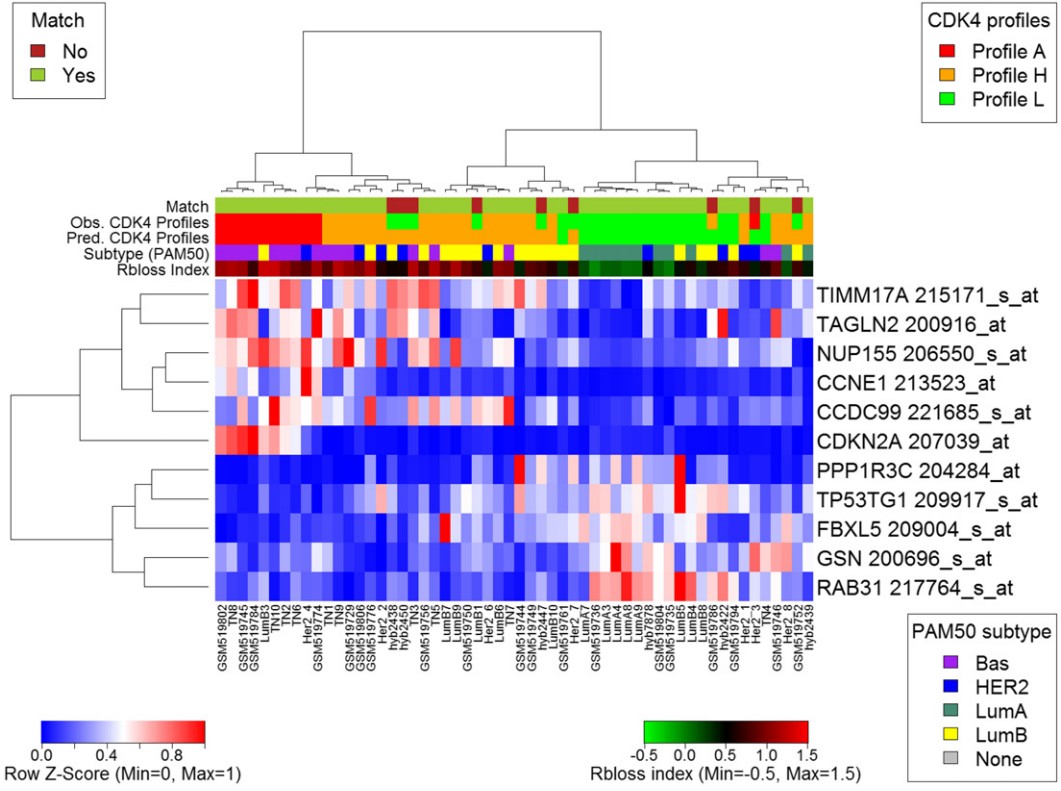

**Figure 3.  An optimized 11-probe gene expression signature predicts the three CDK4 modification profiles in breast tumors.**
RNA was extracted in parallel from the same tumor samples and quantified with the Affymetrix HG-U133plus 2 platform. Heatmaps were drawn with the heatmap.plus R package using the normalized expression values of the optimal 11 probe sets predicting the CDK4 modification profiles corresponding to the 56 breast tumors analyzed in this study. Displayed above each heatmap are the Rb loss index (Ertel *et al*, 2010), the molecular subtype defined with the genefu package based on each sample PAM50 expression value, the observed and predicted CDK4 modification profiles, and the matches between them.

coefficients of correlation to the references of profile L were negatively correlated to these scores and to the Ki-67 index. All the associations described above between the proportions of tumors with the observed CDK4 modification profiles and ER, PR, HER2, or triple-negative statuses, molecular subtypes, histological grade, Oncotype DX and GGI risks, and node involvement were preserved when the predicted profiles were used (middle panels in Fig EV1A and B, Datasets EV3 and EV4).

The consistency and prognostic values of these observations were extended to a cohort of 4,034 gene expression profiles from unique patients published in GEO and ArrayExpress (see Dataset EV1 for the summary of the demographic distributions of clinical parameters of these tumors and Dataset EV8 for detailed clinical records). The classification of the tumors in three clusters (Fig EV4A) and the association of the predicted CDK4 profiles with clinical parameters were similar to those observed in our exploratory cohort (lower panels in Fig EV1A and B, Datasets EV3 and EV4). In these 4,034 patients, CDK4 profile A was predicted in 70% of triple-negative tumors, 18% of HER2-positive tumors, and 5% of ER-positive tumors (lower panels in Fig EV1A and B). In contrast, profile L tumors were enriched in low-risk categories (Oncotype DX score) or grade (GGI) (Fig EV1B). The relapse-free probability was lower in tumors with profile A or H than in those with profile L (Fig EV4B).

## The presence or absence of phosphorylated CDK4 correctly predicts the sensitivity to PD0332991 in 20 breast cancer cell lines

Next, to relate the CDK4 modification profiles to PD0332991 sensitivity in breast cancer cell lines, we selected 20 previously studied cell lines (Finn *et al*, 2009; Barretina *et al*, 2012; Garnett *et al*, 2012). Our selection represents the whole spectrum of PD0332991 sensitivities and includes different breast cancer molecular subtypes, as well as pRb and p16 status (Dataset EV9). We adapted our BrdU incorporation assay (Roger *et al*, 1992) into a 96-well format to directly determine the effect of PD0332991 on the rate of S phase entry (Fig 4A). To compare our data to published results, we completed these measurements with sulforhodamine and MTT assay data (Fig 4C). PD0332991 reduced in a concentration-dependent manner the proportion of DNA-replicating cells in 14 of these 20 cell lines. This confirmed the reported effect of PD0332991 on proliferation (see Fig 4B for representative sensitive and insensitive cell lines and Appendix Fig S5A–E for all the 20 selected cell lines). Only the HER2-positive HCC1954 cell line was sensitive in our hands (Appendix Fig S5A and Dataset EV9), while it is insensitive in Finn's report (Finn *et al*, 2009) and partially sensitive in Kang's work (Kang *et al*, 2014). The lower $IC_{50}$ values (5–50 nM) observed with the BrdU incorporation assay (Dataset EV9) were

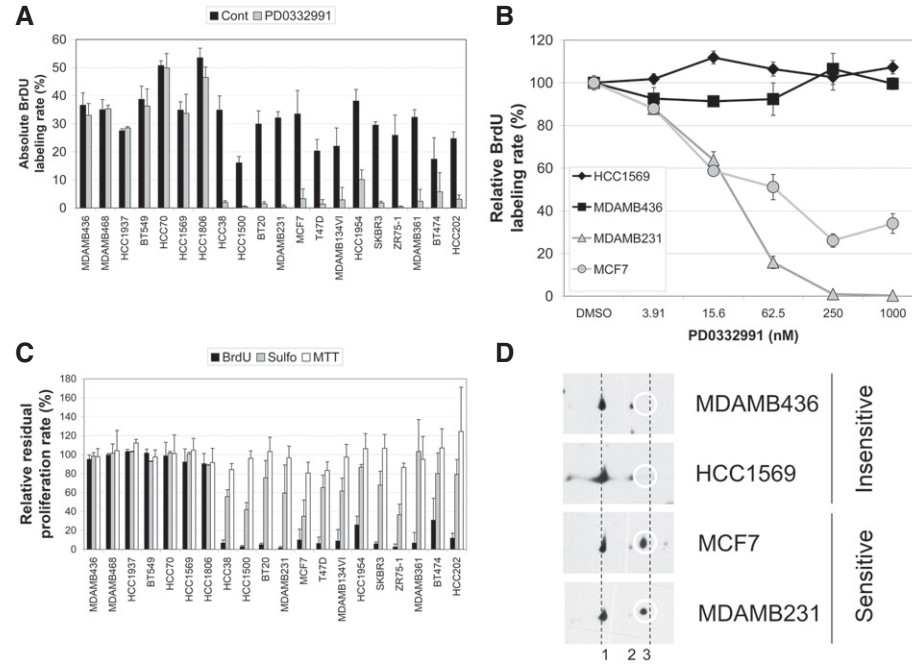

**Figure 4. Sensitivity of 20 breast tumor cell lines to PD0332991.**

A–C  The effect of PD0332991 on the rate of S phase entry was assayed by pulse labeling the indicated cells for 1 h with BrdU as described in the Materials and Methods section. The proportions of BrdU-positive cells in control asynchronous growth in the presence of 10% fetal bovine serum (Cont) or 24 h after challenge with 1 μM PD0332991 displayed in (A) are average values + SD from at least three independent experiments. In (B), typical concentration–action curves of PD0332991 are displayed for two sensitive and two insensitive cell lines. The relative proportion of BrdU-stained cells is expressed as percent of the mean value ± SD of untreated control cells. In (C), the average values + SD of the residual proliferation rate after challenge with 1 μM PD0332991 (expressed as percent of the control value) were determined in at least three independent experiments using BrdU labeling after a 24-h treatment, sulforhodamine assay after a 6-day treatment, and MTT assay after a 48-h treatment.

D  Typical Western blots of 2D-gel electrophoresis profiles of proteins extracted from two sensitive and two insensitive breast cancer cell lines revealed with an anti-CDK4 antibody. The position of the T172-phosphorylated CDK4 form (spot 3) is circled.

close to the PD0332991 IC$_{50}$ on CDK4 activity *in vitro* (11 nM; Toogood *et al*, 2005). The effect of PD0332991 on cell accumulation (sulforhodamine assay) and viability (MTT assay) was generally less marked (Fig 4C, Appendix Fig S5A–E). This indicates that PD0332991 is mainly cytostatic *in vitro*. The reduction of the proportion of cells in the S phase measured in the BrdU assay (Fig 4A–C) was either partial (as in MCF7, HCC1954 and BT474 cells) or complete (as in MDAMB231, ZR75-1, and HCC1500 cells). In HCC1806 cells, this proportion was only slightly reduced to an average of 80% of the control value in five independent experiments (Fig 4C, Appendix Fig S5A). This cell line conserved residual sensitivity to the drug, as confirmed by a marked inhibition of pRb phosphorylations in response to PD0332991 (Fig EV5A). Whole protein extracts of these 20 cell lines in asynchronous growth phase were resolved by 2D-gel electrophoresis and immunodetected by a CDK4

antibody (Fig 4D shows four representative cell lines and Appendix Fig S5A–E all cell lines). The T172-phosphorylated CDK4 form 3 was detected in all sensitive breast cancer cell lines, including the partially sensitive HCC1806 (Fig 5, Appendix Fig S5A). Conceivably due in part to stimulation of their proliferation in the presence of serum (see Fig 1B for MCF7), no cell lines with phosphorylated CDK4 displayed profile L. On the other hand, phosphorylated CDK4 form 3 was undetectable (profile A) in the six cell lines that were completely insensitive to PD0332991 (Fig 5, Appendix Fig S5A–E). Therefore, the presence or absence of the T172-phosphorylated CDK4 form correctly predicted the sensitivity or insensitivity to PD0332991 in these 20 cell lines.

The CDK4 modification profiles of the 20 cell lines were also compared to the immunodetection of key cell cycle regulatory proteins (Fig 5). The relation of the expression of these proteins to

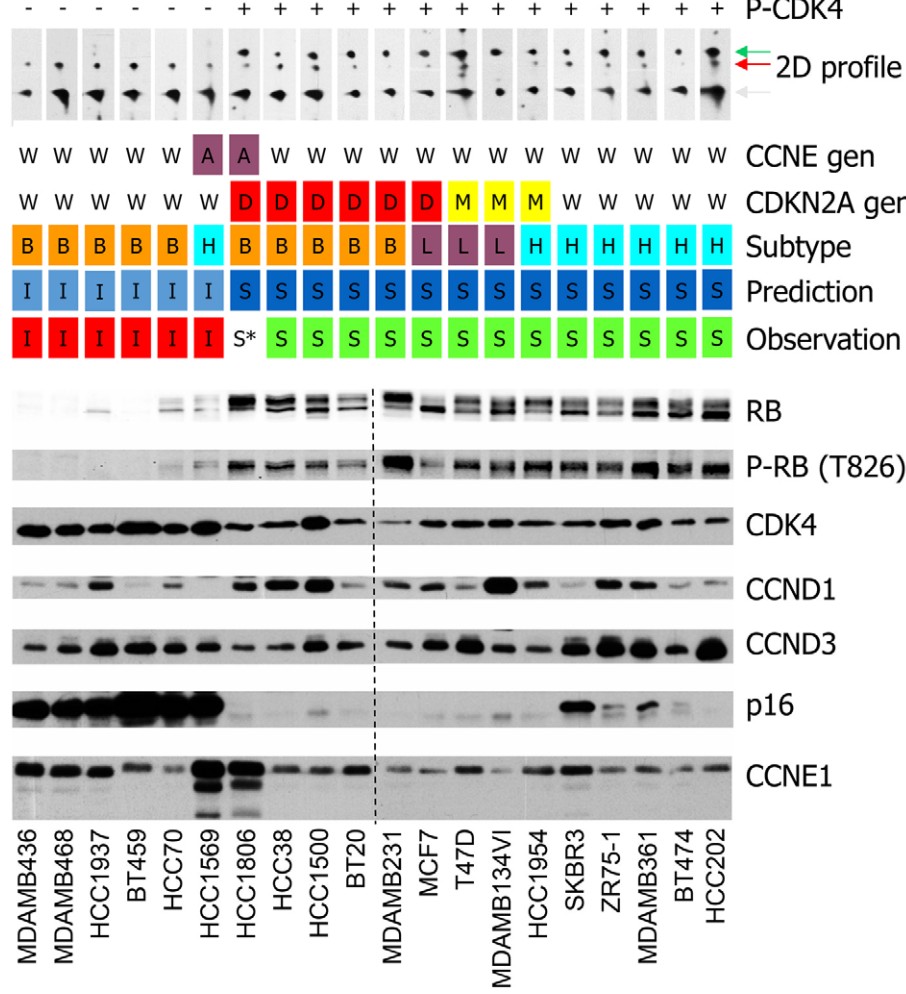

**Figure 5.  Proteomic profiling of breast cancer cell lines shows that the presence of the T172-phosphorylated CDK4 predicts sensitivity to PD0332991.**
Proteins from total extracts were resolved by SDS–PAGE and detected with the indicated antibodies. For each detection, the vertical broken line separates the two blots that were identically processed in parallel. Proteins from total extracts were also resolved by 2D-gel electrophoresis and detected with an anti-CDK4 antibody (the T172-phosphorylated CDK4 is indicated by a green arrow in 2D profile; the non-phosphorylated CDK4 forms corresponding to spots 1 & 2 are indicated by grey and red arrows, respectively). The following are indicated for each cell line: *CDKN2A* and *CCNE1* genomic locus status according to the Cosmic database (W, wild-type; D, homozygous deletion; M, methylation; A, amplification), the corresponding intrinsic molecular subtypes defined by PAM50 gene expression profiles (B, basal-like; H, HER2-positive; L, luminal), and the observed and predicted PD0332991 sensitivity (I, insensitive; S, sensitive).

Source data are available online for this figure.

the sensitivity of the cells to PD0332991 was more ambiguous than was the phosphorylation of CDK4. Indeed, pRb was present not only in all the sensitive cell lines and in HCC1806 cells, but also at lower levels in three of the six insensitive cell lines (HCC1937 and HCC70, which harbor different pRb mutations, and HCC1569). It was undetectable in the other three resistant cell lines, which is consistent with their loss of the *RB1* locus (BT549 and MDAMB468) or with a frame shift mutation of *RB1* (MDAMB436) reported in the Cosmic database (cancer.sanger.-ac.uk; Forbes *et al*, 2015). pRb was phosphorylated on T826 to different extents in all the sensitive cell lines, including the BT20 cell line with the pI388S and pP515L Rb mutations. However, it was also detectable in the insensitive HCC70 and HCC1569 cells and in the partially resistant HCC1806 cells. In the HCC1569 and HCC1806 cell lines, very high levels of cyclin E1 were detected, consistent with amplification of their *CCNE1* genomic locus (Cosmic database). Cyclin D1 levels were lower in most of the insensitive cell lines, except for HCC1937 cells. CDK4 levels tended to be higher in insensitive lines (Fig 5). Finally, in several PD0332991-sensitive cell lines, p16 was undetectable due to deletion of the *CDKN2A* locus (e.g., MCF7) or to its methylation (e.g., T47D and MDAMB134VI; Hui *et al*, 2000). It was detectable in four of the five sensitive cell lines in which the *HER2* locus is amplified (SKBR3, ZR75-1, MDAMB361 and BT474). p16 was strongly elevated in all the insensitive cell lines lacking pRb expression (Fig 5), as reported (Parry *et al*, 1995; Witkiewicz *et al*, 2011). Strikingly, it was also elevated in two cell lines that have detectable pRb phosphorylation (HCC70 and HCC1569). Also remarkably, in the partially resistant HCC1806 cells with strong cyclin E1 expression, p16 was absent due to deletion of the *CDKN2A* locus (Cosmic database). To test whether the probably increased CDK2/1 activity in this cell line can bypass the need for CDK4 activity in initiating DNA synthesis, we treated HCC1806 cells with increasing concentrations of PD0332991 in the presence of R-roscovitine concentrations that block the activity of CDK2 and CDK1 (Meijer *et al*, 1997). As shown in Fig EV5B, roscovitine indeed dose dependently increased the sensitivity of HCC1806 cells to PD0332991. Overall, the presence of T172-phosphorylated CDK4 more accurately predicted the sensitivity to PD0332991 than the expression levels of cell cycle markers.

## Prediction of PD0332991 sensitivity in breast cancer cell lines using gene expression profiles

Next, we explored whether our tumor-based prediction tool can predict CDK4 modification profiles and PD0332991 sensitivity when applied to the gene expression profiles acquired from RNA extracted from our panel of 20 cell lines (accession GSE87006). The distributions of expression values of the 11 genes were generally similar to those observed in the tumors with the corresponding profiles A and H (Appendix Fig S6). The difference reached statistical significance in cell lines *NUP155* and *TAGLN2*, but it was less significant in *CCNE1* ($P = 0.069$) and not significant in *FBXL5* ($P = 0.187$). As for the tumors, the hierarchical clustering of the 20 cell lines based on the expression levels of the 11 probes was consistent with their observed CDK4 proteomic profiles (Appendix Fig S7). Moreover, the predicted and observed CDK4 modification profiles were perfectly concordant (Appendix Fig S7).

As the predicted profile perfectly matched the sensitivity of these 20 cell lines to PD0332991, we extended the validation of our prediction tool to the published gene expression data of breast cancer cell lines that had been acquired with the Affymetrix HG-U133 plus2 platform (which we used to profile our samples). The profiles of 52 cell lines, including the 20 cell lines analyzed in this work, were available. We compared the predicted CDK4 modification profile to the reported PD0332991 sensitivity of the cell lines analyzed by Kang with BrdU labeling (Kang *et al*, 2014), then of the cell lines analyzed by Finn (Finn *et al*, 2009), and finally of the cell lines analyzed at the Broad Institute or Sanger Institute (Barretina *et al*, 2012; Garnett *et al*, 2012). As summarized in Fig 6, the concordance rates between the observed and predicted CDK4 profiles ranged between 88 and 95% in the five studies that had at least 10 cell lines in common with those we used (top 5 rows of the figure). The concordance rates between the predicted CDK4 profiles and the observed PD0332991 sensitivity ranged between 88.2 and 100% in nine studies that had analyzed at least 10 cell lines. The consensus of predicted profiles matched the observed sensitivity in all the cell lines except for SKBR3, AU565 (established from the same patient as SKBR3), DU4475, and CAL120 (concordance rate of 92.3% in the 52 cell lines). Nevertheless, the CDK4 profile and PD0332991 sensitivity were correctly predicted in our transcriptomic profile of SKBR3 (GSE87006; Fig 6) and in one of three transcriptomic profiles (GSE36133) in AU565 (Fig 6). The prediction was generally robust even if the data came from different sources. Indeed, in 45 cell lines out of 52 (86.5%) all predictions converged to a single profile typical of the respective cell lines.

As the published sensitivity diverged from the predicted CDK4 modification profiles in the cell lines DU4475 and CAL120 and was uncertain in CAL85-1, EVSAT and HDQP1, we re-analyzed the effect of PD0332991 on the S phase entry in these models (Fig EV5C). We determined in parallel their CDK4 modification profiles by 2D-gel electrophoresis and the expression of key cell cycle proteins (Fig EV5D). PD0332991 reduced DNA synthesis in EVSAT and CAL120 cells. The other three cell lines were confirmed to be insensitive to the drug, and no pRb was detected in them. This was consistent with the reported loss of *RB1* locus in DU4475 and HDQP1 cells (Robinson *et al*, 2013) and with its mutation in CAL85-1 cells as described in the canSAR database (Tym *et al*, 2016). The predicted presence or absence of CDK4 phosphorylation was consistent with its detection in these cell lines except for CAL120. In this partially sensitive cell line, unlike the prediction, CDK4 phosphorylation was clearly detected but with profile L. pRb was normally present and phosphorylated, but p16 levels were as elevated as in most insensitive pRb-deficient cells (Fig EV5D). This high p16 level was associated with stronger CDK4 expression, consistent with the *CDK4* amplification reported in the Cosmic database. This unexpectedly strong *CDKN2A* expression had likely affected the final prediction in CAL120 cells. Interestingly, a high transcript level of *CDKN2A* was consistently observed in all the tumor cell lines with *CDK4* amplification and high levels of *CDK4* transcript reported in Cosmic database. The opposite was observed in DU4475 cells, in which phosphorylation of CDK4 was correctly predicted by our gene expression signature, but it was associated with pRb loss and total insensitivity to PD0332991 (Fig EV5). In contrast to all the other pRb-deficient cells that we analyzed, p16 was undetectable in this cell line (Fig EV5D), which is consistent with the methylation status

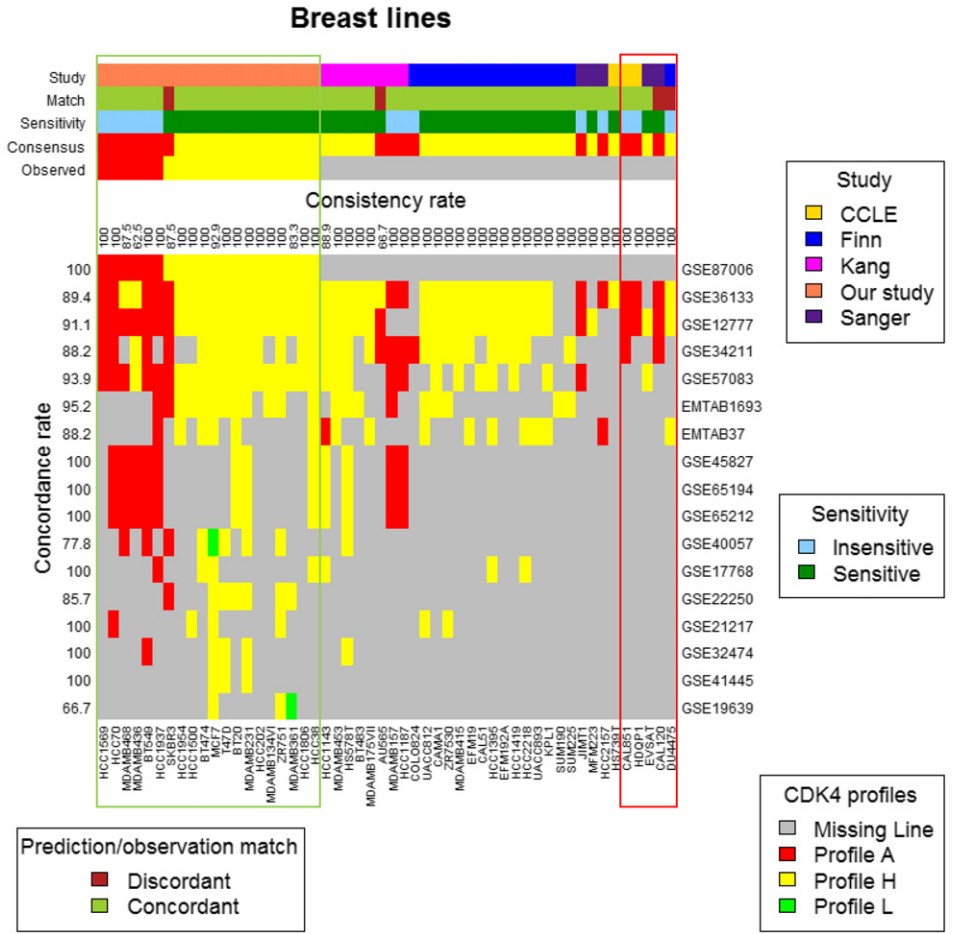

**Figure 6. The 11-probe gene expression predictor predicts PD0332991 sensitivity in 52 breast tumor cell lines.**

The CDK4 modification profiles were predicted in 52 cell lines with known PD0332991 sensitivity by comparing the gene expression of the 11 selected probes to the three representative reference centroids. These expression values were extracted from 17 published studies in which the Affymetrix HG-U133 plus2 platform was used (accession numbers are in the rightmost column). The different studies reporting the effect of PD0332991 on growth or cell cycle entry are color-coded in the first row. The concordance (Match) between the most frequently predicted CDK4 profile for each cell line (Consensus) and PD0332991 sensitivity is color-coded in the second row. PD0332991 sensitivity is color-coded in the third row. This parameter was defined based on the published data, except for the first 20 cell lines (green box) and DU4475, CAL120, CAL85-1, HDQP1, and EVSAT cells (red box), for which our own BrdU labeling data were used. The last two rows of the upper part of the figure illustrate the most frequently predicted CDK4 modification profile (consensus) and the observed CDK4 profile. At the top of the chart (consistency rate), we report the proportion of CDK4 modification profiles that were consistent with the most frequently predicted CDK4 profile. The leftmost column (concordance rate) reports the proportion of predicted CDK4 modification profiles that were concordant with sensitivity of the cell line to PD0332991.

of its *CDKN2A* promoter (Hui *et al*, 2000). Overall, the only confirmed discrepancies between the predicted CDK4 modification profiles and PD0332991 sensitivity in the 52 cell lines were found in the cell lines CAL120 and DU4475. These discrepancies could have been due to a particular molecular alteration (amplification of *CDK4* locus leading to high p16 accumulation) or to the exceptional combination of *RB1* locus loss and *CDKN2A* promoter methylation.

Finally, we examined in the TCGA breast data set the frequency of the molecular defects or their combinations that compromised the prediction of the CDK4 modification profile or its association with PD0332991 sensitivity in the cell lines. Amplification of *CCNE1* associated with strong *CCNE1* expression was identified in 33 out of 1,075 tumors (3.1%). Only one tumor with *CCNE1* amplification also had a *CDKN2A* locus deletion associated with weak expression of *CDKN2A* (0.1%), as observed in HCC1806 cells. Amplification of *CDK4* associated with strong *CDK4* expression was seen in 14

tumors (1.3%), but only eight of them displayed strong expression of *CDKN2A* (0.7%), as observed in CAL120 cells. Finally, out of 655 tumors with available mRNA expression, copy number variation, and mutation data, only two tumors had either a deletion or truncating mutation of *RB1* and a low expression of *CDKN2A* (0.3%) as observed in DU4475 cells.

# Discussion

Palbociclib (PD0332991) is the first CDK4/6 inhibitor approved by the FDA and EMA for treatment of advanced ER-positive breast cancer. Although positive estrogen receptor status was chosen as the only clinical criterion to select eligible patients, a better biomarker is desirable. Indeed, a substantial proportion of patients rapidly progressed upon treatment in the PALOMA-3 study

(Cristofanilli *et al*, 2016), and most HER2-positive and some triple-negative cell lines are also sensitive to palbociclib (Finn *et al*, 2009; Ertel *et al*, 2010). Based on the study of cancer cell lines, loss of pRb function by gene deletion or mutation, amplification of *CCNE1*, and high p16 expression were often associated with resistance to the drug (Dean *et al*, 2010; Ertel *et al*, 2010; Asghar *et al*, 2015). On the other hand, amplification of *CCND1* was associated with sensitivity (Finn *et al*, 2009). Nevertheless, none of these markers alone could unequivocally predict the response to palbociclib in all cases (Finn *et al*, 2015). Our proteomic and genomic data confirmed this conclusion and extended it to gene expression-based indexes, including risk and grade prediction scores (GGI, Oncotype DX) and RB LOH or loss predictors. Like other kinases, the activity of CDK4 requires its phosphorylation in the activation segment (T-loop) (Kato *et al*, 1994a,b). CDK4-activating T172 phosphorylation is the rate-limiting event directly regulated by mitogenic and oncogenic cascades (Bockstaele *et al*, 2006b; Paternot *et al*, 2010; Bisteau *et al*, 2013). By contrast, because CDK6 lacks the critical proline that uniquely follows the phosphorylated T172 in CDK4, the activating phosphorylation (T177) of CDK6 is either undetectable or weak and constitutive (Bockstaele *et al*, 2009). For this reason, and because no T177-phosphospecific CDK6 antibody is currently available, we focused on the analysis of the CDK4 activation. This is the first study to evaluate in tumors and their cell line models the modification profile of CDK4 as a biomarker of the presence of active CDK4 and thus of potential sensitivity to palbociclib.

For the first time, and unexpectedly, we observed that the T172-phosphorylated CDK4 form was absent in a subset of rapidly proliferating tumors and that it was also absent in the PD0332991-resistant cell lines. As T172 phosphorylation is required for the opening of the catalytic cleft of CDK4 and its activity, these observations illustrate that engagement of breast tumor cells in the cell cycle can be dependent on or independent of CDK4 activity, as foreseen by previous cell cycle models (Alevizopoulos *et al*, 1997; Lukas *et al*, 1997). Therefore, the absence of CDK4 phosphorylation seems to be the most general and direct biochemical marker of insensitivity to CDK4/6 inhibitors. In this study, we were also surprised by the high variability of the relative proportion of the T172-phosphorylated form of CDK4 in breast tumors. The distinction between tumors with high or low relative abundance of the phosphorylated CDK4 (profile H and profile L, respectively) proved to be relevant both biologically and clinically: Profile L is strongly enriched in low and intermediate risk and grade tumors, less proliferative tumors, and luminal A tumors.

It is noteworthy that other proposed biomarkers, including detection of pRb (Knudsen & Wang, 2010; DeMichele *et al*, 2015) or its phosphorylation, may not predict palbociclib sensitivity as well as the CDK4 modification profile. The pRb protein was detected in three of six insensitive cell lines, and it was phosphorylated in two of them. Indeed, pRb can be phosphorylated by other kinases, including CDK2 (e.g., in HCC1569 cell line lacking CDK4 phosphorylation). Moreover, mutation of pRb does not always preclude its phosphorylation (Otterson *et al*, 1999), as observed here in BT20 and HCC70 cell lines. On the other hand, we confirmed in several cell lines the association between strong p16 (*CDKN2A*) expression, pRb inactivation, and insensitivity to PD0332991 (Finn *et al*, 2009; Ertel *et al*, 2010). However, high or moderate p16 levels and *CDKN2A* expression were also observed in palbociclib-sensitive

CAL120 and SKBR3 cells, whereas p16 was absent in DU4475 and HCC1806 cells that were completely or partially resistant to PD0332991.

Though 2D-gel electrophoresis is reproducible and sensitive, it is hardly applicable to routine clinical work because it requires frozen tumor samples. On the other hand, phosphorylation events are poorly preserved during formalin fixation. For this reason, because high-performance phosphospecific CDK4 antibodies are yet to be developed, and because CDK4 is expressed at low level, the immunological detection of CDK4 phosphorylation will be hardly adapted to FFPE tumor samples. Those limitations led us to assess whether a surrogate quantitative estimator of CDK4 modifications can be developed based on the CDK4 profiles H and L and phosphorylation-negative profile A (defined by 2D-gel separation) as categorical variables. In this respect, the prediction of these three tumor CDK4 modification profiles based on the gene expression profile of 11 genes is a key promising achievement of our work. An independent validation of this prediction tool with tumors of treated patients is unfortunately not possible yet, because frozen sample collection and gene expression profiling were not included in the protocol of the published palbociclib clinical trials (Finn *et al*, 2015, 2016; Cristofanilli *et al*, 2016). Nevertheless, the following arguments support the validity of our 11-gene signature as a predictor of the CDK4 profile of tumors and thus their potential sensitivity to CDK4/6 inhibitors. First, the concordance rates achieved with our prediction tool were not obtained with tools built on 1,000 random selections of 11 genes or after 1,000 random permutations of the patient labels (Appendix Fig S2). Second, the associations of the proportions of tumors having the different predicted CDK4 modification profiles with key clinical features were confirmed in an independent cohort of 4,034 breast cancer patients with published gene expression profiles. Third, although the 11-gene signature was developed exclusively using the gene expression data of tumors, it predicted the observed CDK4 modification profiles in 24 of 25 breast cancer cell lines. Finally, it correctly predicted the observed or reported PD0332991 sensitivity in 49 of 52 breast cancer cell lines (using our own data in SKBR3 cells). The only discrepancies were explained by particular combinations of defects (combined *CCNE1* amplification and *CDKN2A* deletion in partially resistant HCT1806; amplification of *CDK4* locus leading to high p16 expression in sensitive CAL120 cells; combined loss of pRb and p16 expression in the insensitive DU4475 cells). Analysis of the TCGA data indicated that these combinations rarely occur in tumors and may hence have little clinical significance. Nevertheless, future analyses should evaluate whether the occurrence of such combinations of defects might increase in long-term treatments with CDK4/6 inhibitors.

Once adapted to a qPCR assay compatible with FFPE material and validated, our prediction tool might become clinically useful for optimizing the use of CDK4 inhibitors in the treatment of breast cancer. Our observation that phosphorylated CDK4 is the major modified form of CDK4 in a subgroup of basal-like tumors and most HER2-positive breast tumors provides objective arguments for extending the use of CDK4 inhibitors to these ER-negative tumors. Clinical data presented at the last San Antonio Breast Cancer Symposium support this idea, at least for HER2-positive tumors (Clark *et al*, 2017; Gianni *et al*, 2017). Even for ER-positive tumors, identification of patients who would be insensitive to palbociclib-based treatment is necessary to avoid the

ineffective use of expensive treatment and its side effects (mainly fatigue and neutropenia; Hamilton & Infante, 2016). About 20% of the tumors from PALOMA-3 experienced early progression during treatment with palbociclib in combination with letrozole, indicating that they are intrinsically resistant to both drugs. Importantly, in contrast to patients of the PALOMA-1 and PALOMA-2 trials, the patients from the PALOMA-3 trial had been exposed to endocrine therapy (Finn et al, 2015; Cristofanilli et al, 2016). Although CDK4 activity is required for hormone-independent growth of ER-positive breast cancer (Miller et al, 2011), elevated levels of CDKN2A and CCNE1 together with pRb loss (Bosco et al, 2007) also drive resistance to endocrine therapy (Musgrove & Sutherland, 2009). In this context, we observed that after treatment for 90 days with letrozole, 8 of the 56 tumors analyzed in Miller's study (GSE20181; Miller et al, 2012) had switched to profile A. Prediction of the CDK4 modification profile could therefore identify ER-positive tumors that are unlikely to respond to palbociclib due to their acquisition of endocrine resistance via alterations that also affect palbociclib sensitivity.

In view of the efficacy of palbociclib, its inclusion in neo-adjuvant or adjuvant protocols is foreseen. Although the adverse effects of palbociclib were mild and manageable, they led to dose reduction in up to 34% of the cases in the PALOMA-3 study (Cristofanilli et al, 2016). They may also negatively affect long-term treatment compliance. Sparing palbociclib exposure to patients at lower risk of relapse, who are satisfactorily treated by conventional therapies, would thus be desirable. Stratification of patients based on predefined thresholds of the GGI or of risk indexes is now accepted to select patients for chemotherapy (Ignatiadis et al, 2016). Interestingly, the coefficients of correlation to the centroids corresponding to profiles A and H tumors are positively correlated to the Ki-67 labeling index, the GGI and risk index scores of the tumors, and the expression of cell cycle genes. Conversely, the coefficient of correlation to the centroid corresponding to profile L tumors is negatively correlated to these parameters. Most profile L tumors (mostly luminal A) are thus predicted to be low-grade tumors with a low risk of relapse. Therefore, the CDK4 modification predictor might also be used to bring together two pieces of information required to decide on treatment with CDK4/6 inhibitors: the sensitivity of the tumor and whether it is at high risk and therefore requires dedicated drugs. Most conditions driving the sensitivity or resistance of tumors to CDK4 inhibitors as well as their risk profiles are thus captured in a single assay and summarized in a single statistic that may guide the decision to treat with CDK4 inhibitors.

The efficacy and robustness of our prediction tool are probably due to the method used to define the prediction statistic (correlation to references rather than weighted average) and by the composition of the gene list selected by this approach. On the one hand, these genes include CCNE1 and CDKN2A, the expression of which independently influences sensitivity to PD0332991. Very strong CDKN2A expression reflects functional inactivation or loss of pRb (Gil & Peters, 2006; Witkiewicz et al, 2011), whereas deletion of the CDKN2A locus should facilitate the phosphorylation of CDK4. High p16 levels likely impair CDK4 activation by diverting CDK4 from cyclin D-containing complexes (Sherr, 1996; Asghar et al, 2015) in which CDK4 is phosphorylated (Kato et al, 1994b). Indeed, lack of CDK4 phosphorylation was associated with PD0332991 insensitivity

except when p16 expression was lost due to CDKN2A locus deletion (as in HCC1806 cells) or CDKN2A promoter methylation (as in DU4475 cells). However, as phosphorylated CDK4 was also absent in some tumors with strong CCNE1 expression while CDKN2A expression was moderate or weak (Figs 3, EV3, and EV4A), other factors might contribute to the absence of CDK4 phosphorylation. On the other hand, the selected genes also include two subsets with complementary expression either directly or inversely correlated to the Ki-67 labeling index or the expression of proliferation markers. Indeed, complementary expression was noticed for NUP155, CCDC99, TIMM17A, and TAGLN2 in profile A or H tumors and for RAB31, GSN, TP53TG1, FBXL5, and PPP1R3C in profile L tumors. The expression of the first four genes was positively correlated with the expression of proliferation markers such as CCNB1, MCM5, MCM7, or MKI67, whereas the opposite was observed for the five other genes (Appendix Fig S3, Dataset EV7). Functional evidence linked CCDC99 and TAGLN2 to cell cycle execution or control (see Appendix Supplementary Text). Interestingly, TP53TG1 encodes a long non-coding RNA induced in a wild-type TP53-dependent manner by cellular stress but repressed by cancer-specific promoter hypermethylation (Takei et al, 1998; Diaz-Lagares et al, 2016). Among other functions, it reduces the growth capacity of cancer cells by binding the DNA/RNA binding protein YBX1 to prevent its nuclear localization (Diaz-Lagares et al, 2016). Additional work is needed to clarify whether the last nine selected genes are directly involved in control or execution of the cell cycle.

In summary, we report that the presence and relative abundance of T172-phosphorylated CDK4 varied among breast tumors and cell lines. It was associated with PD0332991 sensitivity in cell lines and with key clinical parameters in tumors. Furthermore, we developed a tool for predicting modification of the CDK4 profile that accurately predicted the tumor profile and sensitivity of breast cancer cell lines to PD0332991. After transposition of this tool to a FFPE-compatible qPCR assay, the utility of this tool for improving and extending patient selection has to be evaluated further in retrospective and prospective studies.

# Materials and Methods

The methods and statistical analyses are fully detailed in the Appendix Supplementary Text.

### Cell culture, DNA synthesis, and cell growth assays

Authenticated human breast carcinoma cell lines were obtained directly from ATCC and passaged for fewer than 6 months after receipt.

For the DNA synthesis assay, cells seeded in triplicates in 96-well plates were incubated for at least 16 h to attach and then challenged with the indicated serial dilutions of PD0332991 for 24 h. One hour before fixation with methanol, 100 μM 5-bromo-2′-deoxyuridine (BrdU, Sigma-Aldrich) and 4 μM 5-fluoro-2′-deoxyuridine (FldU, Sigma-Aldrich) were added to the cells. Immunodetection of DNA-incorporated BrdU was as described (Roger et al, 1992). 4′,6-diamidino-2-phenylindole, dilactate (DAPI, Thermo Fisher Scientific) (1 μg/ml) was used as nuclear counterstain, and round coverslips were mounted in each well of the 96-well plate with ProLong Gold

Antifade Mountant (Thermo Fisher Scientific). Images of the culture plates were taken with a Zeiss Axio Observer.Z1 wide-field microscope and analyzed semi-automatically with a custom-made ImageJ macro to determine the proportion of double-labeled cells. The image acquisition and analysis procedures are detailed in the Appendix Supplementary Text.

The effect of PD0332991 on cellular growth was also evaluated using the sulforhodamine B (SRB) assay as described previously (Vichai & Kirtikara, 2006). Cells seeded in 96-well plates were allowed to attach for at least 16 h before being fixed (time 0 point) or incubated with serial concentrations of PD0332991 for 144 h. Protein-bound SRB solubilized with Tris base solution (10 mM, pH 10.5) was quantified fluorometrically in a Tecan GENios microplate reader at excitation and emission wavelengths of 485 nm and 600 nm, respectively.

Alternatively, cells seeded in triplicates in 96-well plates were allowed to attach for at least 16 h before being incubated with serial dilutions of PD0332991 for 48 h. Thiazolyl blue tetrazolium bromide at a final concentration of 0.6 mg/ml (MTT, Sigma-Aldrich) was added to the cell cultures and incubated for 2 h as described (Slater *et al*, 1963). After removal of the medium, accumulated formazan derivatives were solubilized with DMSO for 30 min with shaking and absorbance was measured in a iMark microplate reader (Bio-Rad) at 540 nm.

### Protein analyses

The references and dilutions of the antibodies used in this work are listed in Appendix Table S1. Equal amounts of whole-cell extract proteins were separated by molecular mass and immunodetected. For 2D-gel electrophoresis, cells were lysed in a buffer containing 7 M urea and 2 M thiourea. Frozen tumor slides (5–7 sections of 7 μm per sample) were solubilized in cold 30 mM Tris buffer pH 8.5 containing 7 M urea, 2 M thiourea and 4% CHAPS with continuous vortexing until unfrozen and then kept agitated for 20 min. After centrifugation at 15,700 *g* for 10 min at 4°C, proteins were quantified. An equal volume of 2-D-sample buffer (7 M urea, 2 M thiourea, 2% CHAPS, 0.4% 3–10 Pharmalytes, and 0.4% DTT) was added to samples normalized to 150 μg proteins. Proteins were separated by isoelectric focusing on immobilized linear pH gradient strips (pH 5 to 8, Bio-Rad) before separation by SDS–PAGE and chemiluminescent immunodetection as described (Bockstaele *et al*, 2006b).

ImageJ was used to quantify the volumes of spot 2 and spot 3 corresponding to the two main modified forms of CDK4 from 16-bit scans of the 2D-gel electrophoresis immunoblots. A circle selection matching the largest spot was created to measure the volume of the two spots and to quantify the background from an area without detectable signal. The background-subtracted volume ratio (spot3/spot2) was considered to define the type of CDK4 modification profile (see Dataset EV2 for values). A profile A was attributed to the tumor when this ratio was below 0.1. A profile L was attributed to the tumor if this ratio lied between 0.1 and 0.9, while a profile H was given in the other cases. Chemiluminescence images of the samples from the cohort of the University of Antwerp and from the CAL85-1, CAL120, DU4475, EVSAT, and HDPQ1 cells were acquired with a Vilber-Loumat Solo7S camera and quantified using the Bio1D software.

### Transcriptomic and bioinformatic analyses

RNA was extracted with Trizol from 50 to 80% confluent cells seeded in 10-cm Petri dishes or from tumor samples as described (Dedeurwaerder *et al*, 2011). The microarray experiment, including probe labeling and hybridization on Affymetrix GeneChip [Human Genome U133 Plus 2.0 (HG133plus2)], was performed at the Bordet Institute or at the University of Antwerp as described (Van Laere *et al*, 2007). In addition, we processed the CEL files of tumors analyzed on HG133A or HG133plus2 Affymetrix array platforms and published in GEO or Array Express (accession numbers and clinical records are given in Datasets EV2 and EV8). Expression data were extracted from the CEL files, background-subtracted, normalized, summarized (median polish option) in R using the frozen RMA package (McCall *et al*, 2010), and converted to signal intensity values. Probe sets with differential expression according to the CDK4 modification profiles of the corresponding tumors were selected in R using either the samr or the pROC package. Reference centroids were generated by computing, for each selected probe set, the average expression value in the selected tumors displaying the same CDK4 modification profile. The expression values for a given tumor of each selected probe set were compared by Spearman correlation to the three centroids corresponding to the three CDK4 modification profiles. The predicted profile was the one corresponding to the centroid that best correlated with the tumor expression profile.

Microarray data analyses, the development of the gene expression-based predictor of CDK4 modification profiles, and the assessment of the Rb loss index and of the Rb LOH score are fully detailed in the Appendix Supplementary Text.

### Patient data and study approval

Only untreated tumors with a proportion of tumoral cells of at least 30% and a corresponding good quality RNA sample were included in our study. The tumors were selected to achieve a balanced distribution of the four main breast cancer expression subtypes determined by IHC. The study was approved by the Institut Jules Bordet Ethics Committee (approval number: CE2161) and conformed to the principles set out in the WMA Declaration of Helsinki and the Department of Health and Human Services Belmont Report. Informed consent was obtained from all patients.

### Data availability

The gene expression profiles of the 20 breast cancer cell lines and the 31 new breast tumors characterized at the Jules Bordet Institute are deposited in GEO (accessions GSE87006 and GSE87007, respectively).

**Expanded View** for this article is available online.

### Acknowledgements

The authors would like to thank all the patients who donated samples for research purposes. We thank Vincent Vercruysse, Samira Majjaj, Naima Kheddoumi, Alex Spinette, and Delphine Vincent for technical assistance; Dr. David Venet for helpful statistical advices; Dr. Amin Bredan for text editing; and Prof. Jacques Dumont for helpful discussions. This study was supported by WELBIO (Walloon Excellence in Lifesciences and Biotechnology), the Belgian

## The paper explained

### Problem

The cell cycle is deregulated in all cancers. New drugs including palbociclib that inhibit the activity of cyclin D-CDK4/6—the first CDK complexes to be activated in G1 phase in response to all oncogenic pathways—are emerging as promising anti-cancer therapeutics. However, tools remain critically lacking for diagnosing tumors that depend on CDK4 activity and hence would respond to CDK4/6 inhibitors. In advanced breast cancers, estrogen receptor (ER) positivity is so far the only criterion to inform the use of palbociclib combined with endocrine therapy. However, this biomarker is not fully satisfying because up to 20% of ER-positive tumors progress within the first two months of treatment by palbociclib combined with fulvestrant. Moreover, to extend palbociclib treatment to ER-negative breast cancers, a biomarker is needed to exclude insensitive tumors.

### Results

We previously identified the activating T172 phosphorylation of CDK4 rather than cyclin D expression as the highly regulated, rate-limiting step determining CDK4 activation and activity, pRb inactivation, and cell cycle commitment. This phosphorylation thus signals the presence of active CDK4, which is targeted by inhibitory drugs. Here, for the first time, we examined this phosphorylation in a collection of breast cancer cell lines and human breast tumors. In cell lines, sensitivity to palbociclib was predicted by CDK4 phosphorylation but not by the level of cyclins D or CDK4 or by the absence of p16 or pRb. Using 2D-gel electrophoresis of protein extracts from frozen tumors, we found that the T172-phosphorylated form is the major modified form of CDK4 in most luminal B and HER2-positive tumors. However, 20% of breast cancers, including a majority of basal-like tumors, lack CDK4 phosphorylation despite their high proliferation rate. Finally, we developed a tool based on the expression of 11 genes that robustly and faithfully predicts the profile of CDK4 modifications.

### Impact

Our work provides proof-of-principle that the relative level of phosphorylated CDK4, and an associated gene expression signature can predict sensitivity or insensitivity to CDK4 inhibitors and their potential clinical benefit. By helping to identify patients who could benefit from the use of CDK4 inhibitors, our tool might help optimize the use of these drugs and its extension to adjuvant protocols. In particular, it will help to extend the indication of CDK4 inhibitors to most HER2-positive tumors and a subset of basal-like tumors that are presently not eligible.

Foundation against Cancer (grants 2010-172 to PPR and IS and 2014-130 to PPR), the Fund Doctor J.P. Naets managed by the King Baudouin Foundation, the Fund Paul Genicot, the Fonds de la Recherche Scientifique-FNRS (FRS-FNRS) under Grant No. J.0002.16 to PPR, Télévie and the foundation Les Amis de l'Institut Bordet. CS is supported by the Breast Cancer Research Foundation (BCRF). The CMMI (LT, VK) is supported by the Hainaut Biomed FEDER Program. JMP is a post-doctoral researcher supported by Télévie. PPR and CS are Senior Research Associates at the FRS-FNRS.

## Author contributions

Conception and design: ER, PPR; development of methodology: KC, ER, LT; acquisition of experimental data: KC, JMP, SP, FR, LC, ER, LT; tumor samples and clinical data: LC, FR, MI, DL, SVL, FS, IS; analysis and interpretation of data including statistical analysis, biostatistics, computational analysis: ER, SB, CS, PPR; writing, review, and/or revision of the manuscript: ER, MI, CS, PPR; study supervision: ER, VK, MP, CS, PPR. All authors have seen and approved the manuscript and its contents.

## Conflict of interest

The Université Libre de Bruxelles has filed a patent application related to these data. ER, PPR, KC, SP, JMP, FR, MI, and CS are listed as co-inventors. No other potential conflict of interest is declared.

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
