## [Review Process File · EMBO Molecular Medicine]

CDK4 phosphorylation status and a linked gene expression profile predict sensitivity to palbociclib

Eric Raspé, Katia Coulonval, Jaime M. Pita, Sabine Paternot, Françoise Rothé, Laure Twyffels, Sylvain Brohée, Ligia Craciun, Denis Larsimont, Véronique Kruys, Flavienne Sandras, Isabelle Salmon, Steven Van Laere, Martine Piccart, Michail Ignatiadis, Christos Sotiriou and Pierre P. Roger

Corresponding authors: Eric Raspé, Christos Sotiriou, and Pierre Roger, Université Libre de Bruxelles

Review timeline:	Submission date:	19 September 2016
	Editorial Decision:	02 November 2016
	Revision received:	14 March 2017
	Editorial Decision:	26 April 2017
	Revision received:	05 May 2017
	Accepted:	09 May 2017

Transaction Report:

Editor: Céline Carret

1st Editorial Decision

02 November 2016

Thank you for the submission of your manuscript to EMBO Molecular Medicine. We have now heard back from the three referees whom we asked to evaluate your manuscript. Although the referees find the study to be of potential interest, they also raise a number of concerns that need to be satisfactorily addressed in the next version of your article.

You will see from the comments below that all three referees have overlapping and complementary comments. They all recognise the interest and potentials of the data. However, they also regret the limited evaluation/validation in an appropriate setting, and it sounds to me that this is absolutely necessary for the article to move forward. Furthermore, referees 2 and 3 also suggest addressing specific problems with the classification used and generation of the signature, besides restructuring the study and exploiting the data better.

Given the balance of these evaluations, we feel that we can consider a revision of your manuscript if you can address the issues that have been raised within the space and time constraints outlined below. Please note that it is EMBO Molecular Medicine policy to allow only a single round of revision and that, as acceptance or rejection of the manuscript will depend on another round of review, your responses should be as complete as possible.

Please note that it is EMBO Molecular Medicine policy to allow only a single round of revision and that, as acceptance or rejection of the manuscript will depend on another round of review, your responses should be as complete as possible.

Please read below for important editorial formatting.

I look forward to receiving your revised manuscript.

***** Reviewer's comments *****

Referee #1 (Comments on Novelty/Model System):

The authors have developed an 11-gene signature that MIGHT be predictive for Palbo de novo resistance. While they perform many correlative study from human samples, they should perform an easy stratification exercise to validate their findings. I have detailed how this could be done in the comment's section.

Referee #1 (Remarks):

Raspe' and colleague have identified a very interesting biomarker for de novo resistance to Palbociclib. Considering the potential use of this new drug in many breast cancer patients, this work has clear translational potential. While the study is well designed and clear, I believe the authors should do an extra-effort to validate their signature. As this is impossible in the context of Paloma1 and 3 (since they lack transcriptional data and I assume RNA cannot be requested or extracted from few responsive and resistant patients) I would suggest they use their 11 gene signature towards a larger set of cell lines for which transcriptional profile exist (there are very large cohort with publically available data). They should validate prospectively 20 or so cell lines for real predicted response of resistant to Palbo. This experiment is easy as data are readily available and the BrdU assay is validated. If their biomarker can predict (as know it can only stratify) response/resistance, I believe this manuscript should be published without hesitation.

Referee #2 (Comments on Novelty/Model System):

The clinical relevance of the 11-gene signature is not really tested in tumors. Without some direct test of correlation with sensitivity to palbociclib, the information given in the manuscript lacks real validation (only correlation with Cdk4 T172 phosphorylation is demonstrated).

Referee #2 (Remarks):

The manuscript by Raspe et al. reports the use of Cdk4 T172 phosphorylation as a biomarker to predict response to Cdk4 inhibitors in cancer treatment. The presence of this phosphorylation, detected in 2D gels, correlates with sensitivity to palbociclib, a Cdk4/6 inhibitor approved for the treatment of ER+/HER- breast cancers. This correlation is almost perfect in a panel of 20 breast cancer cell lines. A similar analysis in human tumors suggests the presence of three patterns of Cdk4 phosphorylation named as profile 1-3. Further bioinformatics studies suggest that these profiles correlate with the expression of several transcripts and a signature of 20 transcripts is proposed as the best way to predict Cdk4 phosphorylation profile in human tumors.

In general, this is a very interesting proposal that deserves proper clinical evaluation. There are however a few points that make the proposal a bit weak in some aspects.

1. Whereas the correlation between Cdk4 T172 phosphorylation and sensitivity to palbociclib is very strong (Fig. 1-2), the correlation with other markers seems to be under-interpreted by the authors. In Figure 2, it is quite clear that both Rb and Phospho-Rb levels display a very strong correlation with sensitivity. By using a lower exposure of the same film (WB in Fig. 2), one could simply use total RB levels to predict sensitivity. Or in other words, HCC1937 and perhaps HCC1569 also display low levels of phospho-Cdk4 T172 despite their insensitivity. This is not to say that Cdk4 phosphorylation is a good marker but it seems that the potency of RB levels (and of p16) as a marker is somehow dismissed.
2. In the absence of RB, Cdk4 is not phosphorylated (Fig. 2). The molecular basis of this correlation is not clear as, in principle, one could expect to see tumors with strong Cdk4 activation but insensitive to Cdk4 inhibition because Rb is absent (and the oncogenic event is therefore downstream of Cdk4). One possibility is that pRb deletion leads to p16 overexpression and this event switches Cdk4 off. That would explain Figure 2. However, if this is the case, one could expect that patients with p16 deletion that become resistant to therapies by deleting pRB would be positive for the Cdk4 T172 phosphorylation (as they are p16 null and likely express cyclins) and, yet, they would be insensitive to Cdk4 inhibitors (due to Rb loss). In these patients, the information given by Cdk4 T172 phosphorylation would be misleading.
3. The technical analysis of Cdk phosphorylation using 2D gels is fantastic in the manuscript but it is likely to be difficult to setup. The phospho-specific antibodies used in Fig. S2 seem to recognize Cdk4 T172 with good specificity. It would be a great addition if the authors could demonstrate these Abs working in one-dimension gels to identify Cdk4 phosphorylation in breast cancer cells.
4. The distinction between profile 2 and 3 is very relative and it is very difficult to understand their relevance. It is clear that the correlation with expression profiles is good and is probably better than considering these two profiles as a single unit. If the authors were to use combined profiles 2-3, how the analysis of human tumors change? Is the 11-gene signature still informative?
5. Just looking at the phospho-spots in Fig. 3A one could assume that Profile 2 represents a protein with stronger phosphorylation in T172 (and more similar to the situation in sensitive cell lines) when compared to Profile 3. The authors have decided to use the inverse order probably because the correlation of profile 3 with clinical data is stronger. This issue should be clearly explored and discussed in the manuscript. Do the authors have any idea or hypothesis on the molecular meaning of profile 3 versus 2? Otherwise, it seems that most of the work in the second half of the manuscript is not really based on a solid evaluation of the molecular status of Cdk4.
6. The clinical relevance of the 11-gene signature is not really tested. Without some direct test of correlation with sensitivity to palbociclib, the information given in the manuscript lacks real validation (only correlation with Cdk4 T172 phosphorylation is demonstrated).
7. The reader gets a bit frustrated in the discussion about the genes in the signature. Apart from the usual suspects, CCNE1 and CDKN2A, it seems that the other hits are not very informative. The discussion about TP53TG1 is interesting but one prediction is that one could exchange this transcript by any other p53 target and the signature will still work. Is that testable?
8. Similarly, the discussion of the other 8 transcripts as simply reporting "the proliferation status of the tumor" is very weak. The authors indicate that NUP155, CCDC99, TIMM17A and TAGLN2 correlate with cell cycle genes. However, these are not typically cell cycle genes and if they simply indicate proliferation they could be exchanged by any other cell cycle gene. Is that correct? If not, why the signature selected these 4 genes and not others? Following the same rationale, it does not seem that the other 4 genes, RAB31, GSN, FBXL5 and PPP1R3C typically represent antiproliferative genes.
9. Cyclin E1 can also activate Cdk1 and roscovitine is a very good Cdk inhibitor.
10. The first sentence in the last paragraph in page 21 is an opinion and does not seem to require a citation, unless the authors disagree with it.
11. In Fig. 3A, is the presence of high Ki67 levels a criterion for being a Profile 1 tumor?

Referee #3 (Comments on Novelty/Model System):

The manuscript presents an interesting and timely study aimed at identifying mediators of response to CDK4/6 inhibition in breast cancer, and at the development of biomarkers predictive of that response.

The study presents an extensive study of BRCA cell lines and their CDK4 phosphorylation profiles, and it purports to define "modification profiles" associated w/ response/non-response (or sensitivity/insensitivity) to Palbociclib treatment.

The authors define these profiles in BRCA cell lines, and in multiple cohorts of primary tissues samples, and show a concordance of these profiles between in-vitro and in-vivo models.

The evaluation of the predictive accuracy of their biomarker is sub-optimal, in that they rely only on cross-validation, rather than genuine prediction on an independent test set. This makes the results more likely to be biased (suffering from over-fitting). I recognize the added value of their evaluation on a large compendia of publicly available expression profiles (n=4034), but accuracy in that cohort can only be indirectly evaluated based on the relative distribution of other phenotypes of interest (e.g., pam50, grade, etc.) within the predicted modification profiles, thus making it less compelling.

In addition, the write-up of the manuscript could be improved. At the moment, it is rather verbose and not very structured, reading more like a stream of thoughts than a well organized and tight list of results.

Some specific comments:

* It is not immediately clear why profile 3 (phospho<spot2) should not be sensitive to inhibition as well, given that the phosphorylation site is still present (although lower than spot2). Isn't it also what one should conclude from Figure 2 (top), where all the cell lines with "intermediate" level of the phospho-site turn out to be sensitive? On a related note, the text (in Results) reads "the phosphorylated form was also detected but its abundance was much lower than in breast cancer cell lines, below the threshold mentioned above." There's no mention of this threshold anywhere in the preceding text.

* It would be helpful to actually label the modification profiles w/ meaningful labels (e.g., 1: no-phospho; 2: phospho>spot2; 3: phospho<spot2) so as to help the reader (right now, I found myself often going back and forth to check which was which).

* Regarding the verbosity of the text, the end of a section in Results reads

"Thus, the absence or presence of T172- phosphorylated CDK4 discriminated tumors that are unlikely to respond to CDK4 inhibitors from those that are potentially sensitive."

The preceding text is somewhat convoluted and non clearly organized, thus making the final statement far from self-evident.

* The description of the design of the predictor and its evaluation is not sufficiently clear and precise, while (again) unnecessarily verbose. What are the inputs (genes/proteins)? Was any feature selection performed, or were the 10 or so genes manually selected (based on prior knowledge)?

* The Discussion section is, again, very long and unstructured, making it hard to read.

1st Revision - authors' response

14 March 2017

Answers to Reviewer #1's criticisms and suggestions:

(Modifications to the manuscript are underlined)

"Raspe' and colleague have identified a very interesting biomarker for de novo resistance to Palbociclib. Considering the potential use of this new drug in many breast cancer patients, this

work has clear translational potential. While the study is well designed and clear, I believe the authors should do an extra-effort to validate their signature. As this is impossible in the context of Paloma1 and 3 (since they lack transcriptional data and I assume RNA cannot be requested or extracted from few responsive and resistant patients) I would suggest they use their 11 gene signature towards a larger set of cell lines for which transcriptional profile exist (there are very large cohort with publically available data). They should validate prospectively 20 or so cell lines for real predicted response of resistant to Palbo. This experiment is easy as data are readily available and the BrdU assay is validated. If their biomarker can predict (as know it can only stratify) response/resistance, I believe this manuscript should be published without hesitation.”

We warmly thank the referee for his/her general comment about the interest of our work, his/her suggestion to validate our predictive signature, and his/her recommendation.

Definition of the CDK4 modification profile or the quantification of the expression of the 11 genes selected in our predictive tool require fresh frozen material. Unfortunately, as acknowledged by the referee, such samples were not included in the protocol of the clinical trials set up to test the benefit of palbociclib published to date. Validation of our tool on tumor material from these trials is therefore impossible at the moment. As pointed out by the referee, prediction of the cell line sensitivity to palbociclib may be used as evidence to support the validity of the prediction tool. It is important to clarify that our prediction tool was set up exclusively with tumor data and that prediction of the sensitivity to palbociclib of the 20 cell lines analyzed in our work provided already partial validation clues. As also asked by the Editor and the other referees, the structure of the Result section was completely modified and rewritten to put forward the independence of this validation process. A new sub-section (page 15) exclusively dedicated to the use of the cell lines to validate the prediction tool was added to the Results. To follow the referee’s suggestion, we extended this analysis to 32 other cell lines with published palbociclib sensitivity and gene expression data. Concordance rate between predicted CDK4 profile and observed or reported palbociclib sensitivity in the 52 cell lines analyzed in total was above 90% (new Fig 6). We further characterized experimentally 5 cell lines with discordances between observation and prediction or with uncertain palbociclib sensitivity (new Fig EV5 C,D). In three cases, we were able to solve the discrepancies or to establish the true sensitivity of the cell lines. In two cases, the discrepancies between prediction and observation were confirmed and explained by an infrequent genetic defects or the combination of mutually exclusive alterations (amplification of CDK4 leading to high p16 expression and loss of Rb combined with p16 loss). However, as these alterations or combination of alterations are rare in the breast tumor dataset described in the TCGA (as detailed in the text, page 18), our prediction tool is likely to correctly predict CDK4 modification profile and sensitivity to palbociclib in most tumor samples. This argument together with additional validation clues (comparison to predictions based on random lists of 11 genes or performed after random permutation of patient label, similarity of the relations between CDK4 modification profile prediction and key clinical parameter in an extended cohort of 4034 patients) are now discussed in the Discussion (pages 21,22).

We have much appreciated the comments and suggestions of this Reviewer that allowed us to improve the clarity of the manuscript and strengthen our conclusions. We sincerely hope to have adequately addressed his/her comments and suggestions.

Answers to Reviewer #2’criticisms and suggestions:

(Modifications to the manuscript are underlined)

We thank this reviewer for the excellent summary of our work and for emphasizing its importance and identifying several ways to improve our manuscript (“*In general, this is a very interesting proposal that deserves proper clinical evaluation. There are however a few points that make the proposal a bit weak in some aspects.*”).

Below we fully answer his/her specific suggestions, questions and criticisms :

1. “*Whereas the correlation between Cdk4 T172 phosphorylation and sensitivity to palbociclib is very strong (Fig. 1-2), the correlation with other markers seems to be under-interpreted by the authors. In Figure 2, it is quite clear that both Rb and Phospho-Rb levels display a very strong correlation with sensitivity. By using a lower exposure of the same film (WB in Fig. 2), one could simply use total RB levels to predict sensitivity. Or in other words, HCC1937 and perhaps*

HCC1569 also display low levels of phospho-Cdk4 T172 despite their insensitivity. This is not to say that Cdk4 phosphorylation is a good marker but it seems that the potency of RB levels (and of p16) as a marker is somehow dismissed.”

We agree with the referee that total Rb, phospho-Rb and p16 levels are also often correlated to palbociclib sensitivity or resistance. This is now better emphasized in the manuscript with references. For instance, the group of Knudsen has rightly suggested to use the absence of RB coupled to high p16 IHC to predict in tissue sections the PD0332991 insensitivity of a tumor. However, among 11 tumors lacking CDK4 phosphorylation, two were associated with high CCNE1 expression but normal CDKN2A (p16) level. In cell lines, we now observed that very high p16/CDKN2A expression is also associated with high CDK4 expression due to CDK4 amplification in palbociclib-sensitive cells (CAL120). Conversely p16 is absent in cells that are partially (HCC1806) or totally resistant to palbociclib (DU4475; in this unique case with combined losses of pRb and p16, phospho-CDK4 was also misleading as predicted by the referee). These new data are now illustrated in Fig EV5 A,B) and described in the text (page 17).

Concerning the lower level of Rb in three insensitive cell lines, reducing the exposure time as suggested by the referee could also lead to the wrong appreciation that Rb level is low in at least two sensitive cells (BT20 and ZR75-1). This could lead to a wrong prediction of their palbociclib sensitivity status. On the opposite, we could also have used a higher exposure of the same film to emphasize the presence of total pRb and its phosphorylation in the three insensitive cell lines leading to the wrong prediction that they are sensitive. The lower abundance of pRb in these lines was apparent only because several cell lines were compared in a same blot. In IHC detections, such side by side comparison is more challenging experimentally. So, using only the presence of pRb as single marker to decide whether a tumor would benefit or not from a treatment with palbociclib will probably lead to an unacceptable rate of diagnosis errors. High rates of diagnosis error may also be expected with detection of only phosphorylated Rb by considering the slow migrating pRb band or direct detection with a phospho-Rb-specific antibody. Indeed, these levels were also quite variable among the sensitive cell lines, especially in the sensitive MCF7 cells. Furthermore, mutations of Rb do not always affect its phosphorylation (such as in the BT20 and HCC70 cell lines). Its phosphorylation depends not only on CDK4/6 but also on CDK2. In the case of over-activated CDK2 (for example owing to amplification of CCNE1), the presence of phosphorylated Rb will not necessarily signal the sensitivity to palbociclib. Finally some mutations affect expression of Rb, as in the CAL85-1 cells, but this is not always the case. In all cases, exceptions are more frequent with cell cycle markers than with CDK4 T172-phosphorylation. Our main message is that the presence of CDK4 T172-phosphorylation is more robustly associated to palbociclib sensitivity than other cell cycle markers especially if they are considered individually. Detection of phosphorylated epitopes can also be challenging when FFPE material is used. Phosphorylation events are often labile and lost during the time of tissue fixation. A more robust assay compatible with the use of FFPE samples would thus be desirable. We hope that our tool, once transposed to qRT-PCR, will reach this goal. This is now discussed (pages 20, 21).

2. “In the absence of RB, Cdk4 is not phosphorylated (Fig. 2). The molecular basis of this correlation is not clear as, in principle, one could expect to see tumors with strong Cdk4 activation but insensitive to Cdk4 inhibition because Rb is absent (and the oncogenic event is therefore downstream of Cdk4). One possibility is that pRb deletion leads to p16 overexpression and this event switches Cdk4 off. That would explain Figure 2. However, if this is the case, one could expect that patients with p16 deletion that become resistant to therapies by deleting pRB would be positive for the Cdk4 T172 phosphorylation (as they are p16 null and likely express cyclins) and, yet, they would be insensitive to Cdk4 inhibitors (due to Rb loss). In these patients, the information given by Cdk4 T172 phosphorylation would be misleading.”

The Referee is perfectly right! To check this possibility, we found one cell line (DU4475) that combines these mutually exclusive (as generally accepted) losses of pRb and p16 owing to deletion of Rb and methylation of the CDKN2A promoter. In this unique cell line, insensitivity to palbociclib indeed occurred without complete loss of CDK4 phosphorylation. These new data are now shown in Fig. EV5C,D and discussed in the text (page 17). This situation is rare in the breast tumors described in the TCGA study, as detailed in the text page 18. Nevertheless, we agree with the referee that the frequency of this combination might become higher in case of acquired resistance to palbociclib. The following sentence has been added page 18: “Nevertheless, future analyses should evaluate

whether the occurrence of such combination of defects might increase in long term treatments with CDK4/6 inhibitors.

3. *“The technical analysis of Cdk phosphorylation using 2D gels is fantastic in the manuscript but it is likely to be difficult to setup. The phospho-specific antibodies used in Fig. S2 seem to recognize Cdk4 T172 with good specificity. It would be a great addition if the authors could demonstrate these Abs working in one-dimension gels to identify Cdk4 phosphorylation in breast cancer cells.”*

The use of an antibody to detect phosphorylated CDK4 in cell lines and tumors for diagnostic purpose is appealing but unfortunately limited in practice. No performant phospho-CDK4 antibody are available (some commercial polyclonals have appeared recently but we doubt of their specificity as they use as positive controls HeLa cells that lack CDK4 phosphorylation because of high p16 induced by HPV E7-mediated inactivation of pRb!). The polyclonal antibody preparation used for the former figure S2 (now Fig 1A,C) was provided to us in 2004 by Cell Signaling Technology. They were thinking it was inactive but they lacked a positive control. We fully characterized this trial antibody as published in Bockstaele Mol Cell Biol 2006. This antibody is very phosphospecific on CDK4 but rather ‘dirty’, generally requiring a purification or separation of CDK4 (by IP and/or 2D electrophoresis). Nevertheless, it could be used (it was critical) in the study by Fisher group published in Mol Cell (Merzel-Schachter et al 2013). This preparation has never been commercialized and the bulk of its stock was unfortunately lost at CST. They were unable to repeat it (one production was sold but it was inactive and removed from the catalog after 6 months). We keep preciously the few microliters left as reference material.

4. *“The distinction between profile 2 and 3 is very relative and it is very difficult to understand their relevance. It is clear that the correlation with expression profiles is good and is probably better than considering these two profiles as a single unit. If the authors were to use combined profiles 2-3, how the analysis of human tumors change? Is the 11-gene signature still informative?”*

We agree with the referee that the distinction between profile 2 and 3 (now called H (high) and L (low)) is relative. At least in part, these two categories likely reflect a continuum. This continuum probably reflects at least in part the relative intensity of the oncogenic stimulation of CDK4 phosphorylation and thus the resulting rate of tumor cell proliferation (as suggested by the regulation of the CDK4 phosphorylation level upon mitogenic stimulation of MCF7 cells described in the new figure EV1B). This distinction was admittedly empirical at the beginning of the study. Somehow, it can be compared with the Ki-67 based distinction between luminal A and B subtypes (with which profiles L and H correlate). The Spot3/Spot2 ratio used to distinguish the profile H and L correlates well with the RbLOH score. On the opposite, this RbLOH score cannot distinguish the profile A (‘absent’, the former profile 1) from the profile H. One important observation of our work is that the tumors with low levels of phosphorylated CDK4 are mostly luminal A tumors with low risk. The coefficient of correlation to the centroids corresponding to the three CDK4 modification profiles are indeed positively or negatively correlated to several risk scores (Fig 2A, Fig EV1B). The latter values are also continuous. However, thresholds have now been validated to attribute risk categories according to the level of these scores reached in tumors. In analogy with the use of classical chemotherapy, knowing the risk of relapse of the tumor may also help to establish the balance between clinical benefit and exposure to adverse effects of treatment with palbociclib (i.e. to decide or not to treat). Therefore, implementation of our tool indicates with one statistic not only whether a tumor will be sensitive or not to palbociclib but also whether the balance between clinical benefit and adverse events will be favorable. This might be especially useful when palbociclib will be included in adjuvant or neo-adjuvant protocols. This is detailed in the discussion in the paragraph on the clinical utility of our tool (page 23).

Distinction of tumors based only on whether CDK4 phosphorylation is present or not as suggested is of course possible but less informative (something like only predicting the ER status). In practice, this classification strategy means to use only the coefficient of correlation to the centroid corresponding to tumors devoid of CDK4 phosphorylation in a binary A/non-A comparison, for example. A ROC analysis comparing the coefficient of correlation to the reference corresponding to tumors without detectable phosphorylated CDK4 (profile A) with the binary category A/non-A yielded an area under the curve of 0.95. With an optimal threshold of the coefficient of correlation to the profile A reference set at 0.868, a specificity of 100%, a sensitivity of 90.9% and an accuracy of 98% were achieved in the 56 tumor data set. However, in cell lines, the same analysis yielded an

AUC of 70.2, a specificity of only 50%, a sensitivity of 100% and an accuracy of 85% with a threshold set at 0.818. Actually, three cell lines without detectable phosphorylated CDK4 (BT549, MDAMB436 and HCC1569) presented lower correlation coefficients for all three references while the correlation to profile A reference was still correctly the highest. By contrast, the coefficient of correlation to the reference corresponding to tumors wherein phosphorylated CDK4 is detectable but low (profile L) was lower in absolute term in all profile A cell lines compared to profile H ones. Hence, using this coefficient of correlation improved their classification (specificity of 100%, sensitivity of 92.9% and an accuracy of 95% with a threshold set at 0.455). In the particular case of these three cell lines, using only one coefficient of correlation lead to a wrong prediction while taking them all into account corrected the prediction. Taking into account the correlation to the 3 centroids thus reinforces the predictive reliability of our tool and is probably central to its efficacy.

5. *“Just looking at the phospho-spots in Fig. 3A one could assume that Profile 2 represents a protein with stronger phosphorylation in T172 (and more similar to the situation in sensitive cell lines) when compared to Profile 3. The authors have decided to use the inverse order probably because the correlation of profile 3 with clinical data is stronger. This issue should be clearly explored and discussed in the manuscript. Do the authors have any idea or hypothesis on the molecular meaning of profile 3 versus 2? Otherwise, it seems that most of the work in the second half of the manuscript is not really based on a solid evaluation of the molecular status of Cdk4.”*

The original order of former Fig3A (now Fig 1D) was arbitrary and chosen mainly to highlight the paradoxical lack of CDK4 phosphorylation in some highly proliferative tumors. This order has been changed in the new Fig 1D as we realized from the referee's comment that this order may not seem logical.

Regarding the molecular meaning, it most likely reflects the intensity of the oncogenic stimulus that induces the CDK4 phosphorylation. The profile L is typical of a normal stimulated cell in culture like diploid fibroblasts (for instance, please see in Bockstaele MCB 2006) or thyroid epithelial cells that we studied a lot. By contrast the profile H is typical of most tumor cell lines with CDK4 phosphorylation (as observed in this study). CDK4 phosphorylation is the last step of a complex dynamic process requiring the association of a cyclin D with CDK4 (which is favored by CDKN1 encoded proteins (p21, p27) and opposed by CDKN2 encoded proteins (p16, p15..)), proper phosphorylation of p21 or p27, and the action of regulated activating kinase(s) to counterbalance the rapid dephosphorylation of CDK4 by still unknown phosphatases. Therefore, the profile of CDK4 should integrate various parameters and be sensitive to each of them. We previously characterized (Bockstaele et al MCB 2006) that the phosphorylated form 3 of CDK4 is enriched in cyclin D-CDK4 complexes, whereas the opposite is observed for the form 2 which is present in both 'free' CDK4 and p16-bound CDK4. Although this dichotomy is not absolute, the distinction between profile L and profile H should also reflect in part the ratios between the relative expression of CDK4, cyclins D, p16 and p21 or p27. It is very interesting (and unexpected) that this distinction correlates with, and actually predicts, the relapse risk of a tumor.

6. *“The clinical relevance of the 11-gene signature is not really tested. Without some direct test of correlation with sensitivity to palbociclib, the information given in the manuscript lacks real validation (only correlation with Cdk4 T172 phosphorylation is demonstrated).”*

This is right and now more clearly stated in the manuscript (page 21 and in the conclusion of the Discussion). As also acknowledged by Referee 1, independent validation of our prediction by comparison with palbociclib sensitivity in patients is not possible for the moment because the biological material required for it was not collected in the palbociclib trials. As suggested by Referee 1, we include in the revised version of the manuscript a detailed analysis of the prediction of the palbociclib sensitivity of an enlarged collection of breast cancer cell lines. As the prediction tool was exclusively developed based on tumor data, the good concordance rate between predicted CDK4 modification profile and observed palbociclib sensitivity (48 out of 52 cell lines) provides a first line of independent validation of the tool. Other arguments include the worst performances of tools based upon random lists of 11 genes, the worst performances of the tool applied after 1000 random permutation of the patient labels and the similar associations of the proportions of tumors displaying the 3 different CDK4 modification profiles with classical clinical parameters in an extended cohort of 4034 tumors with published gene expression data. The different validation arguments are now explicitly addressed in the Discussion page 21.

7. *“The reader gets a bit frustrated in the discussion about the genes in the signature. Apart from the usual suspects, CCNE1 and CDKN2A, it seems that the other hits are not very informative. The discussion about TP53TG1 is interesting but one prediction is that one could exchange this transcript by any other p53 target and the signature will still work. Is that testable?”*

It is interesting that the ‘usual suspects’ were captured by our statistical approach among other genes. This suggests that our approach, and the use of the CDK4 modification profiles as references, were really meaningful, biologically and clinically. However, these two genes were not sufficient for the prediction and a combination of other genes was required. It is difficult to estimate now to what extent the other selected genes are actively and functionally involved in the phenotype or are just markers statistically co-expressed with other genes playing a more direct role in the tumor phenotype. Further functional analyses should address such questions. Nevertheless, these genes can be separated in two groups with opposite relations to proliferation markers. This opposite behavior is probably a critical key to the classification performance of our tool. A plethora of genes follow the same behavior. However, because our prediction is based on a Spearman correlation, the relative rank of the expression values of these genes need at least to be similar to the relative rank of the genes they would replace. It is thus possible that some genes directly or indirectly related to the cell cycle progression can be interchanged with others but this will not be possible in all cases. This is illustrated below with the particular case of TP53TG1 and its replacement with other genes controlled by p53.

The p53 target gene expression profile is strongly influenced by the cellular context (in particular to the mutation landscape of the tumor). Therefore, the possibility to exchange TP53TG1 in the signature by any other p53 target might be difficult to evaluate. Nevertheless, we verified if TP53TG1 could be replaced by CDKN1A (p21), a classical p53 target with a key role in the control of CDK4 activity. The equivalence of two probe sets in predicting CDK4 modification profile imposes two constraints. First, the relative levels of the two probe sets needs to be correlated between the samples. This seems to be the case for CDKN1A. Its expression is positively correlated with the expression of TP53TG1 in cell lines (slope 0.93, Rsquare 0.248, pvalue 0.025) and tends to do so in tumors (slope 1.25, Rsquare 0.067, pvalue 0.054). Second, the relative rank of the probe sets need to be similar among the other probe sets. This implies that the expression values of the two probe sets need to be similar. However, as the expression level of CDKN1A is higher than the one of TP53TG1, the second condition for equivalence is not met. This could explain why the performance of the classification was worst when TP53TG1 is replaced by CDKN1A. A recent publication of Estreller's group (Diaz-Lagares et al, 2016, Proc Natl Acad Sci USA, 113(47):E7535-E7544) showed that its expression depends on the hyper-methylation status of a CpG island located in its promoter detectable in up to 26% of gastric tumors analyzed. This hyper-methylation of the TP53TG1 promoter was associated with a worst outcome of the disease. However, how and why the TP53TG1 promoter was methylated in cell lines and tumors was not addressed in their work. Because the regulation of the expression of TP53TG1 seems more complex than described before and may depend on other factors than p53, we removed the quote to a possible link with p53 in the discussion of the article. Nevertheless, the inverse relation between expression of TP53TG1 and proliferation marker seems to be physiologically relevant as these authors showed that overexpression of TP53TG1 in cells lacking it reduced their proliferation capacity. This new information has been added in the revised version of our manuscript (page 25).

8. *“Similarly, the discussion of the other 8 transcripts as simply reporting “the proliferation status of the tumor” is very weak. The authors indicate that NUP155, CCDC99, TIMM17A and TAGLN2 correlate with cell cycle genes. However, these are not typically cell cycle genes and if they simply indicate proliferation they could be exchanged by any other cell cycle gene. Is that correct? If not, why the signature selected these 4 genes and not others? Following the same rationale, it does not seem that the other 4 genes, RAB31, GSN, FBXL5 and PPP1R3C typically represent antiproliferative genes.”*

We now show the correlation of the 11 genes with the Ki-67 index of the tumors in addition to the correlation with MKI67 gene expression in the Appendix Fig S3. Our proposal that the expression of 8 of the 11 genes selected in our tool report the proliferation status is based on the observed strong direct or inverse correlation between the expression of these genes and the Ki-67 labeling index or of the expression of key cell cycle markers in tumors. Whether these genes are actively involved in the execution of the cell cycle or whether they are just markers co-regulated with such

genes is still an open question. We also tested whether the relationship with cell cycle markers observed in tumors was reproduced in cell lines. The positive correlation to cell cycle markers was similar in tumor and cell lines for CCDC99, TAGLN2, and NUP155. This correlation was significant in the cell lines only for TAGLN2, and NUP155. The negative correlation to cell cycle markers was similar in tumor and cell lines for GSN, FBXL5 and TP53TG1. This correlation was significant in the cell lines only for TP53TG1. For PPP1R3C, TIMM17A and RAB31, opposite relations were observed between tumors and cell lines. This relation was significant in the cell lines only for TIMM17A. Since the relations between the expression of RAB31, TIMM17A, GSN, FBXL5 and PPP1R3C and proliferation markers in the cell lines is not significant or even opposite to what is seen in tumors, the relations observed in the tumors between these genes and markers of proliferation may be indirect and/or the consequence of a tumor-stroma crosstalk. To avoid confusion and unnecessary lengthening of the article, we included the sentence "Additional work will be needed to clarify whether the last nine selected genes are directly involved in control or execution of the cell cycle." at the end of the corresponding paragraph page 25. The known function of the different genes is described in the Appendix. A direct link to proliferation can however be expected for at least 4 genes among those with similar relation to cell cycle markers in cell lines and tumors. The link of CCDC99 also called SPDL1 or Spindly with cell cycle is obvious as this gene codes for a protein required for efficient chromosome spindle formation and mitotic checkpoint regulation (Barisic and Geley, 2011, Cell cycle, 10:449-456). The role of TAGLN2 in cell cycle is less obvious although its silencing was reported to significantly inhibit cell proliferation and invasion in HNSCC cells (Nohata et al, 2011, Oncotarget 2:29-42). NUP155 is a nucleoporin part of the nucleopore complex (Imamoto and Funakoshi, 2012, Current Opinion in Cell Biol 24:453-459). At the onset of mitosis, the nuclear envelope and the nuclear pore complex disassemble. However, nucleoporins often remain associated in sub-complexes associated with mitotic structures, such as kinetochores or the spindle. In particular, Nup188, a component of the complex wherein Nup155 is located localizes to spindle poles during mitosis (Itoh et al, Cancer Sci 2013; 104: 871-879). Whether Nup155 may be required for that function or could play another role in M phase is worth exploring. Severe DNA segregation defects are indeed observed after depletion of Nup155 in *C. elegans* (Franz et al EMBO Journal 2005, 24:3519-3531). The case of TP53TG1 is discussed above. Therefore, at least 4 key genes of the list seem directly linked positively or negatively to the control or the execution of the cell cycle.

9. *"Cyclin E1 can also activate Cdk1 and roscovitine is a very good Cdk inhibitor."*

This is perfectly right, as first shown by the group of Kaldis to our knowledge (Aleem et al, 2005). CDK2 or CDK1 (CDK2/1) is now mentioned in the text. However activation of CDK1 by cyclin E1 might require specific conditions like the absence of p27 (Aleem 2005). As roscovitine indeed similarly inhibits both CDK2 and CDK1 (now mentioned in the text, ref Meijer 1997), whether cyclin E1 acts on CDK2 or CDK1 does not influence our conclusions in HCC1806 cells.

10. *"The first sentence in the last paragraph in page 21 is an opinion and does not seem to require a citation, unless the authors disagree with it."*

This sentence was removed.

11. *"In Fig. 3A, is the presence of high Ki67 levels a criterion for being a Profile 1 tumor?"*

In tumor profile 1 (now called A) a paradoxical lack of CDK4 phosphorylation is observed despite a high Ki67 index. This high Ki67 index distinguishes them from normal breast. However, the Ki67 index was not initially used as a criterion to define the different CDK4 profiles. The text was rewritten and clarified. The association of the different profiles with Ki-67 index is now shown in Fig 1E.

We have much appreciated the very meaningful comments and questions of this referee. We sincerely hope to have adequately addressed them and succeeded to clarify our message and conclusions.

Answers to Reviewer #3's criticisms and suggestions:

We thank this reviewer for the excellent summary of our work and for emphasizing its interest. The various weaknesses that he/she rightly identified have been corrected. The manuscript has been re-structured and rewritten following his/her suggestions.

Below we fully answer his/her comments and suggestions and we detail the important modifications to the manuscript (modifications are underlined).

General comments:

“The evaluation of the predictive accuracy of their biomarker is sub-optimal, in that they rely only on cross-validation, rather than genuine prediction on an independent test set. This makes the results more likely to be biased (suffering from over-fitting).”

“In addition, the write-up of the manuscript could be improved. At the moment, it is rather verbose and not very structured, reading more like a stream of thoughts than a well organized and tight list of results.”

We indeed realized that the order of presentation and discussion of the results was suboptimal and could be misleading. We initially started with the description of the cell lines data to link this work to our previous ones. The drawback of this choice is that the reader could be confused on the use or not of cell line data in the set up of our prediction tool. **This tool was based exclusively on tumor data.** We therefore started now with the description of the tumor data and of the set up of the prediction tool. This was next completed with the relation between CDK4 modification profile and cell line palbociclib sensitivity and finally with an explicit paragraph describing the **independent validation** of our prediction tool based on cell line data. The structure of the introduction was modified to increase its logic, starting from general data on the cell cycle control, applying them to the breast cancer and finally to the particular case of palbociclib sensitivity. The write up of the discussion was also revised. We started from the question, state of the art and the way to answer it. We followed by highlighting the remarkable character of our main observation and its implication. Next we discussed the validity and the utility of our prediction tool and finally discuss some mechanistic aspects of our observations at the basis of the performance of our prediction tool. Regarding the style, we now use as much as possible shorter sentences to avoid verbosity. Furthermore, description of the figures is now limited to the most salient information.

Specific comments:

“It would be helpful to actually label the modification profiles w/ meaningful labels (e.g., 1: no-phospho; 2: phospho>spot2; 3: phospho<spot2) so as to help the reader (right now, I found myself often going back and forth to check which was which).”

Labels have been changed to A (absent: no-phospho), H (high: phospho \geq spot2) or L (low: phospho<spot2). We hope that these more meaningful and explicit labels will help remembering the associated phenotype.

“It is not immediately clear why profile 3 (phospho<spot2) should not be sensitive to inhibition as well, given that the phosphorylation site is still present (although lower than spot2). Isn't it also what one should conclude from Figure 2 (top), where all the cell lines with "intermediate" level of the phospho-site turn out to be sensitive?”

The tumors with profile 3 are indeed expected to be sensitive. We agree with the referee that the distinction between profile 2 and 3 (now called H and L) is relative. Please note that the two common prognostic indexes, the GGI and the Oncotype DX score are also continuous parameters. Application of validated thresholds to these indexes allows to determine if a tumor is or is not at risk of relapse. Their use is now validated to decide to treat or not a tumor with classical chemotherapy. Owing to its impressive efficacy, extension of the use of palbociclib to adjuvant or neo-adjuvant protocols is expected. In this case, estimating the risk of relapse is of great importance to avoid unnecessary lifelong exposure to the mild but significant side effects of the drug and to avoid unaffordable cost increase to the social security systems. As the coefficients of correlation between the tumor gene expression profiles and the 3 centroids representative of the tumors with the 3 CDK4 modification profiles are positively or negatively correlated to the GGI or Oncotype DX scores, these coefficients can also be used to estimate the risk of relapse of a tumor. In contrast to profile H and profile A tumors, profile L tumors are found to be at low risk of relapse. Those data are now

illustrated in Fig 2, Fig EV1B and Fig EV4B, Appendix Fig S4). Application of our tool will therefore provide two vital information required to decide whether to include or not palbociclib in an adjuvant or neo-adjuvant protocol (is the tumor sensitive, is the tumor at risk). This is discussed page 23. Because tumors with low CDK4 phosphorylation level are associated with a low risk of relapse, the benefit from the treatment for these patients might be smaller than the adverse effect burden.

“On a related note, the text (in Results) reads “the phosphorylated form was also detected but its abundance was much lower than in breast cancer cell lines, below the threshold mentioned above.” There’s no mention of this threshold anywhere in the preceding text.”

Spot3/Spot2 ratio thresholds used to classify the tumors and cell lines are explicitly included in the text and discussed.

ROC analysis showed that the empirical threshold of the Spot3/spot2 ratio set at 0.9 defined tumors with a positive GGI risk status with the highest combined accuracy (0.59) and positive predictive value (0.95). On the other hand, the empirical threshold of the Spot3/spot2 ratio set to 0.1 was the lowest which distinguished sensitive and insensitive cell lines with an accuracy of 100%.

“Regarding the verbosity of the text, the end of a section in Results reads ... The preceding text is somewhat convoluted and non clearly organized, thus making the final statement far from self-evident.”

This section of the text has been completely rewritten and restructured as explained above. Throughout the text, verbosity was reduced by trying to replace (as much as possible) long sentences by shorter ones. Furthermore, description of the figures was reduced to emphasize the most salient observations.

“The description of the design of the predictor and its evaluation is not sufficiently clear and precise, while (again) unnecessarily verbose. What are the inputs (genes/proteins)? Was any feature selection performed, or were the 10 or so genes manually selected (based on prior knowledge)?”

Sorry for being unclear. **The 11 genes were not selected manually, but by a biostatistical process.** The description of the design of the predictor was clarified and extended in the main text of the article. It is detailed in the Appendix. CDK4 modification profiles analyzed by 2D gel electrophoresis were used as categorical variables to identify genes differentially expressed among different tumors using the Statistical Analysis of Microarray tool implemented in the sam package of R. Alternatively, genes associated with the phenotype categories after binarization were selected by Receptor/Operator analysis with the pROC package in R. Criteria used to select genes are described in the Appendix. Selections of genes were based on randomly selected subsets of tumors data wherein the proportions of A, H, and L tumors were kept constant. This led to the selection of about 700 lists of genes. Next a reference for each profile (named centroid) was built by computing the mean of each gene in the list among the representative tumors having the corresponding profile. The gene expression levels of the selected genes in a tumor were next compared to the three references corresponding to the three profiles. The predicted profile is the profile corresponding to the reference leading to the highest Spearman coefficient of correlation. Performance of the prediction tools built based on the different lists of selected genes were evaluated by comparing the concordance rate of predicted and observed profile in the complementary subset of patients used to select the genes. The number of probe sets in the gene list was optimized in a stepwise manner by removing one at a time each probe set of the list.

Also to avoid confusion, the cell line data description used for the independent validation of the predictor was moved after the tumor data and prediction tool description. We extended the validation by the analysis of an additional set of 32 cell lines with published palbociclib sensitivity and gene expression data. The CDK4 modification-based 11-gene signature indeed correctly predicted the reported or observed palbociclib sensitivity. Concordance rate between predicted CDK4 profile and observed or reported palbociclib sensitivity in the 52 cell lines analyzed in total was above 90% (new Fig 6). We further characterized experimentally 5 cell lines with discordances between observation and prediction or with uncertain palbociclib sensitivity (new Fig EV5 C,D). In three cases, we were able to solve the discrepancies or to establish the true sensitivity of the cell lines.

“The Discussion section is, again, very long and unstructured, making it hard to read.”

The discussion was completely re-written and reorganized as described above.

We have much appreciated the comments and suggestions of this Reviewer that allowed us to improve the clarity of the manuscript and strengthen our conclusions. We sincerely hope to have adequately addressed his/her comments and criticisms.

2nd Editorial Decision

26 April 2017

Thank you for the submission of your revised manuscript to EMBO Molecular Medicine. We have now received the enclosed reports from the referees that were asked to re-assess it. As you will see the reviewers are now globally supportive and I am pleased to inform you that we will be able to accept your manuscript pending the following final amendments:

1) Please address the minor text changes commented by referees 2 and 3. Please provide a letter INCLUDING the reviewer's reports and your detailed responses to their comments (as Word file).

2) in M&M, patients data: please include there a statement that the experiments conformed to the principles set out in the WMA Declaration of Helsinki and the Department of Health and Human Services Belmont Report.

Please submit your revised manuscript within two weeks. I look forward to seeing a revised form of your manuscript as soon as possible.

***** Reviewer's comments *****

Referee #1 (Remarks):

all issues were correctly addressed by the authors

Referee #2 (Comments on Novelty/Model System):

Medical impact may be high but it is difficult to tell as validation is still preclinical in the manuscript.

Referee #2 (Remarks):

The revised manuscript by Raspé et al is much improved compared with the original submission, especially in the clarity of the presentation of the results, including the re-organization affecting the independent validation generated in cell lines. In general, the use of T172 phosphorylation to indicate Cdk4 activity is more than a solid concept very well described in the text. Owing to the technical problems of using this signal in the Clinic, the identification of the gene signature correlating with this mark is a reasonable strategy that is, in general, well executed in the manuscript.

The authors have made an impressive effort to clarify all the previous obscure aspects in the text and they have honestly discussed the pros and cons of the strategy. I agree with most answers to referee's questions and with the changes in the text. My one two additional comments are the following:

After this manuscript is published it is likely that investigators and clinicians will try to study Cdk4 T174 phosphorylation using reported antibodies. It would be very important to clarify somewhere in the text (or supplementary text) that available antibodies don't anymore as it was explained to the reviewer.

I still think that the conclusion that "Most conditions driving the sensitivity or resistance of tumors to CDK4 inhibitors [...] are thus captured in a single assay" is an over-conclusion. The authors have

no clue about the meaningfulness of most of these markers in the context of Cdk4 activity and their link to cell cycle regulation is a bit forced as discussed. E.g. classifying a nuclear pore protein as a gene linked to cell cycle control simply because is a nuclear pore protein is not appropriate in this context. Even in the most obvious example, Spindly, there is no known reason why expression of the corresponding transcript (apparently not reported as an E2F target) may correlate with Cdk4 activity or sensitivity to Cdk4 inhibitors. Considering that these are the most obvious examples of correlation with cell cycle I would stick with the conclusion that additional is needed to clarify the functional relevance of the 9-gene signature with Cdk4 activity.

Referee #3 (Comments on Novelty/Model System):

The authors adequately addressed the reviewer's criticisms.

Referee #3 (Remarks):

The authors adequately addressed the reviewer's criticisms, and this reviewer feels the manuscript is suitable for publication as is.

Minor edits/clarifications:

In Section "Phosphorylated CDK4 predicts PD0332991 ..." (page 12), please specify in the opening paragraph the prediction performance on the 20 cell lines (e.g., n/m cell lines were correctly predicted as sensitive ...).

2nd Revision - authors' response

05 May 2017

We warmly thank all reviewers for their comments and recommendation regarding our revised manuscript EMM-2016-07084-V2 entitled "CDK4 phosphorylation status and a linked gene expression profile predict sensitivity to Palbociclib".

All the minor changes they suggested were done in the revised (V3) manuscript.

Referee #1 (Remarks):

all issues were correctly addressed by the authors

Referee #2 (Comments on Novelty/Model System):

Medical impact may be high but it is difficult to tell as validation is still preclinical in the manuscript.

Referee #2 (Remarks):

The revised manuscript by Raspé et al is much improved compared with the original submission, especially in the clarity of the presentation of the results, including the re-organization affecting the independent validation generated in cell lines. In general, the use of T172 phosphorylation to indicate Cdk4 activity is more than a solid concept very well described in the text. Owing to the technical problems of using this signal in the Clinic, the identification of the gene signature correlating with this mark is a reasonable strategy that is, in general, well executed in the manuscript.

The authors have made an impressive effort to clarify all the previous obscure aspects in the text and they have honestly discussed the pros and cons of the strategy. I agree with most answers to referee's questions and with the changes in the text. My one two additional comments are the following:

After this manuscript is published it is likely that investigators and clinicians will try to study Cdk4 T174 phosphorylation using reported antibodies. It would be very important to clarify somewhere in the text (or supplementary text) that available antibodies don't anymore as it was explained to the reviewer.

In the antibody table (now Appendix Supplementary Table S1), we have completed the note concerning the T172-phosphospecific CDK4 antibody :
 « * sample of a noncommercialized phosphospecific CDK4 (Thr172) antibody produced by immunizing rabbits with a keyhole limpet hemocyanin-coupled peptide antigen to T172-phosphorylated human CDK4 and purified by protein A- and immunogen-based affinity column separation. See Bockstaele et al Mol. Cell. Biol. 26,5070 (2006) for characterization. This antibody is very phosphospecific on CDK4 but generally requires a prior purification or separation of CDK4 (by IP and/or 2D electrophoresis). Nevertheless, it could be used (it was critical) in a study by Robert Fisher group (Merzel-Schachter et al Mol. Cell 50,250 (2013)). This preparation has never been commercialized and the bulk of its stock was unfortunately lost at CST. To our knowledge, CST was unable to reproduce it until now (one production was sold but it was inactive and removed from the catalog after 6 months). We preciously keep the few microliters left as reference material. »

I still think that the conclusion that "Most conditions driving the sensitivity or resistance of tumors to CDK4 inhibitors [...] are thus captured in a single assay" is an over-conclusion. The authors have no clue about the meaningfulness of most of these markers in the context of Cdk4 activity and their link to cell cycle regulation is a bit forced as discussed. E.g. classifying a nuclear pore protein as a gene linked to cell cycle control simply because is a nuclear pore protein is not appropriate in this context. Even in the most obvious example, Spindly, there is no known reason why expression of the corresponding transcript (apparently not reported as an E2F target) may correlate with Cdk4 activity or sensitivity to Cdk4 inhibitors. Considering that these are the most obvious examples of correlation with cell cycle I would stick with the conclusion that additional is needed to clarify the functional relevance of the 9-gene signature with Cdk4 activity.

The concept of a single statistic to represent most conditions driving the sensitivity or resistance of a tumor to CDK4 inhibitors is derived from the efficient clustering of the tumors using the collective expression values of the 11 selected genes as illustrated by the heatmaps presented in the article and the relations between the expression of proliferation markers or predictive indexes and the correlation coefficients obtained by comparing tumors gene expression profiles with the 3 reference profiles described in the supplementary material. We agree with the referee 2 that additional work outside the scope of this article will be required to understand better the relations of the individual genes of the signature with the control or the execution of the cell cycle. To avoid unnecessary confusion, the sentence « Most conditions driving the sensitivity or resistance of tumors to CDK4 inhibitors... » has been removed. We also removed in this paragraph the notion that the nucleoporin Nup155 might be a gene associated with cell cycle execution (despite our observation that its transcript level significantly correlates with KI-67 and cell cycle markers).

Referee #3 (Comments on Novelty/Model System):

The authors adequately addressed the reviewer's criticisms.

Referee #3 (Remarks):

The authors adequately addressed the reviewer's criticisms, and this reviewer feels the manuscript is suitable for publication as is.

Minor edits/clarifications:

In Section "Phosphorylated CDK4 predicts PD0332991 ..." (page 12), please specify in the opening paragraph the prediction performance on the 20 cell lines (e.g., n/m cell lines were correctly predicted as sensitive ...).

Thank you for this comment. The conclusion of this section indeed should to be more immediately apparent.

The subheading of this section has been changed in « The presence or absence of phosphorylated CDK4 correctly predict the sensitivity to PD0332991 in 20 breast cancer cell lines. » Moreover, we now end this paragraph by the sentence « Therefore, the presence or absence of the T172-phosphorylated CDK4 form correctly predicted the sensitivity or insensitivity to PD0332991 in these 20 cell lines. »

Corresponding Author Name: Pierre P. ROGER

Manuscript Number: EMM-2016-07084_V2